# Single test-based diagnosis of multiple cancer types using Exosome-SERS-AI for early stage cancers

Hyunku Shin [1,10], Byeong Hyeon Choi [2,3,10], On Shim[1], Jihee Kim[1], Yong Park[4], Suk Ki Cho[5], Hyun Koo Kim [2,6] ✉ & Yeonho Choi [1,7,8,9] ✉

Early cancer detection has significant clinical value, but there remains no single method that can comprehensively identify multiple types of early-stage cancer. Here, we report the diagnostic accuracy of simultaneous detection of 6 types of early-stage cancers (lung, breast, colon, liver, pancreas, and stomach) by analyzing surface-enhanced Raman spectroscopy profiles of exosomes using artificial intelligence in a retrospective study design. It includes classification models that recognize signal patterns of plasma exosomes to identify both their presence and tissues of origin. Using 520 test samples, our system identified cancer presence with an area under the curve value of 0.970. Moreover, the system classified the tumor organ type of 278 early-stage cancer patients with a mean area under the curve of 0.945. The final integrated decision model showed a sensitivity of 90.2% at a specificity of 94.4% while predicting the tumor organ of 72% of positive patients. Since our method utilizes a non-specific analysis of Raman signatures, its diagnostic scope could potentially be expanded to include other diseases.

Cancer remains the leading cause of death in modern societies, but early diagnosis improves cancer outcomes by providing timely and optimal treatment[1–3]. Thus, early-stage cancer detection by in vitro diagnosis is an important goal in the biomedical field, allowing routine cancer management[4]. Currently, several cancer biomarkers, including carcinoembryonic antigen (CEA)[5] and prostate-specific antigen (PSA)[6], have been targeted in diagnostic and prognostic strategies, but these are rarely present at earlier stages of malignancy. Other cancer types still have no effective prescreening tools[7]. Recently, biomarkers, including CTC[8], cfDNA[9], and extracellular vesicles (EV)[10,11] carrying

cytosolic and membrane information from tumor cells, have emerged as attractive biomedical targets in liquid biopsy approaches for many cancer types[12].

Exosomes, which are a subtype of nano-sized small EVs, are enclosed by a lipid bilayer and actively secreted from living cells, so they can be utilized to acquire information about cancer cells noninvasvely[13,14]. In addition to their direct secretion by tumor cells, various biological factors, including immune regulation[15], tumor microenvironment[16], and angiogenesis[17], are associated with the exosomes and potentially induce compositional differences in blood[18,19].

[1]EXoPERT Corporation, Seoul 02580, Republic of Korea. [2]Department of Thoracic and Cardiovascular Surgery, College of Medicine, Korea University Guro Hospital, Seoul 08308, Republic of Korea. [3]Korea Artificial Organ Center, Korea University, Seoul 02841, Republic of Korea. [4]Division of Hematology-Oncology, Department of Internal Medicine, Korea University College of Medicine, Seoul 02841, Republic of Korea. [5]Division of Thoracic Surgery, Department of Thoracic and Cardiovascular Surgery, Seoul National University Bundang Hospital, Seongnam 13620, Republic of Korea. [6]Department of Biomedical Sciences, College of Medicine, Korea University, 02841 Seoul, Republic of Korea. [7]School of Biomedical Engineering, Korea University, Seoul 02841, Republic of Korea. [8]Department of Biomedical Engineering, Korea University, Seoul 02841, Republic of Korea. [9]Interdisciplinary Program in Precision Public Health, Korea University, 02841 Seoul, Republic of Korea. [10]These authors contributed equally: Hyunku Shin, Byeong Hyeon Choi. ✉e-mail: kimhyunkoo@korea.ac.kr; yeonhochoi@korea.ac.kr

Accordingly, if the cancer-related characteristic patterns of exosomes are discriminated between cancer types, simultaneous liquid biopsy for multiple cancer types would be possible.

In this context, vibrational spectroscopy techniques that obtain information on vibrational and rotational modes of the chemical structure are emerging as major tools to identify different types of bio-samples[20]. In particular, Raman spectroscopy is a powerful method for detecting compositional differences in biomaterials such as exosomes based on the advantages of being simple, non-destructive, and requiring less amount of analytes[21–24]. With the development of plasmonic enhancing methods, for example, surface-enhanced Raman spectroscopy (SERS), many groups have used Raman spectroscopy to analyze EVs[20,22,25–30]. These studies demonstrate that EVs and exosomes from various biosamples from cell-cultured media[27,31] to blood[32,33] can be identified without labeling and specific antigens. In particular, current advances in multi-variate statistical methods and machine-learning technologies, including artificial intelligence (AI), make identifying these vibrational spectra more precise and easy[20,31,33,34].

Here, we demonstrate a liquid biopsy method that combines AI and SERS to simultaneously diagnose multiple cancer types by label-free analysis of plasma exosomes (Fig. 1a). Our method acquires SERS signals of isolated exosomes, then analyzes them with deep learning models. There are two outputs: cancer diagnosis and tissue of origin (TOO) discrimination. (Fig. 1b) In the first step, the deep learning model classifies each signal as normal or cancerous, yielding a score of cancer presence. In the second step, multiple classifier models trained cancer types using the one-vs.-rest method generate TOO determinations of positive predictions from the first step. In this paper, we demonstrate the diagnostic performance of this system using 520 test samples that had not been used for training. The samples include six cancer types (lung, breast, colon, liver, pancreas, and stomach) and early-stage cancer patients.

## Results

### Exosome isolation and signal measurement

Exosome purity and surface chemical status are significant factors in our label-free detection because other biomolecules and chemicals on the exosome surface affect the resulting Raman signals. Accordingly, we employed size exclusion chromatography (SEC) that isolates exosomes based on the hydrodynamic size of vesicles[35,36]. Since SEC does not use additional chemical reagents that produce undesired Raman signals in the isolation process, the disturbance of signals can be minimized in label-free SERS detection. Plasma samples from 210 healthy controls (HC) and 543 cancer patients had pathologically confirmed their diagnoses by each medical center and stored by routine protocols in each medical center.

We flowed the provided 500-μL plasma sample through the SEC column to obtain fractions according to particle size. To determine fractions as an exosome suspension, exosome protein expression and physical properties such as size, concentration, and morphology were analyzed. By western blotting of proteins from each eluted fraction, we identified exosome markers, including CD9, CD63, CD81, and TSG101 at a certain fraction range (Supplementary Fig. 1). A mixture of fractions lacking interfering impurities, such as lipoprotein, calnexin, and soluble proteins, were subjected to subsequent experiments as an exosome suspension. We confirmed that the exosome markers CD9, CD63, and CD81 were present in both HC and patient samples (Fig. 2a). Vesicle-like particles were observed using cryo-transmission electron microscopy (cryo-TEM) (Fig. 2b). Nanoparticle tracking analysis (NTA) showed particle sizes of 100–150 nm at $10^9$–$10^{10}$ particles/mL (Fig. 2c, d).

An Au nanoparticle (AuNP)-aggregated array chip for SERS was prepared using centrifugation-based sedimentation methods that we have previously reported[37] (Supplementary Fig. 2). After colloidal AuNPs were precipitated, NPs were applied to the APTES-functionalized glass surface as 2.5-mm diameter dots (Fig. 2e). One chip is designed to have ten detection spots to increase detection

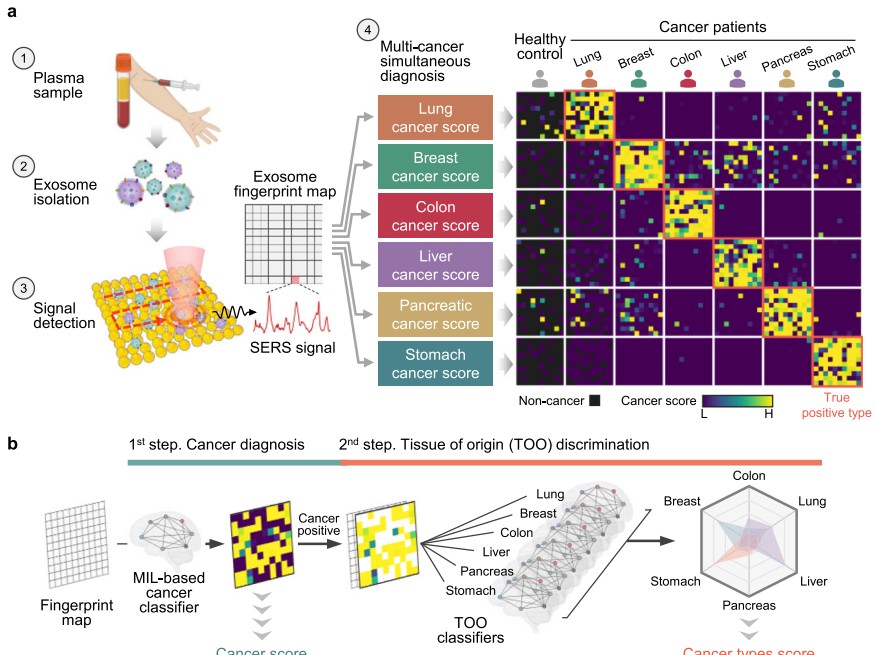

**Fig. 1 | One test-multi cancer using exosome-SERS-AI. a** Overview. Exosome suspension is dropped onto an Au nanoparticle-aggregated array chip and thoroughly dried. Signals were observed at 100 spots (10 × 10) per sample and analyzed by AI algorithms. The system outputs predictions about cancer presence and tissue of origin. A heat map shows actual examples of the representative predicted results for each cancer status. **b** AI framework. In the first step, diagnostic scores are assigned as the mean values of the multiple instance learning (MIL)-based cancer classifier results. In the second step, signals predicted by the previous cancer classifier are analyzed, then an average score is calculated using six types of prediction models. Cartoons were created with BioRender.com.

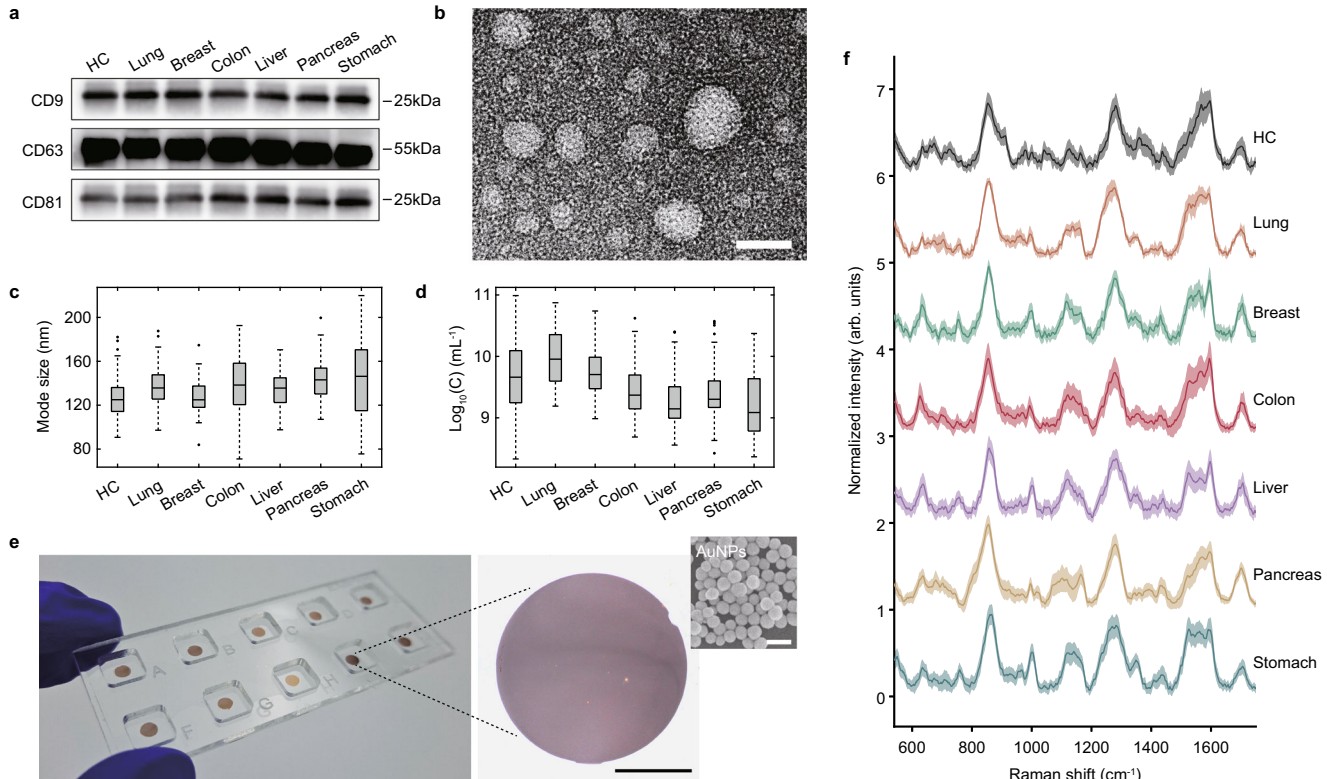

**Fig. 2 | Exosome isolation and detection. a–d** Evaluation of isolated exosomes. **a** Western blot showing exosome marker expression. **b** Cryo-TEM image to show the morphology of isolated vesicles. Scale bar, 50 nm. **c, d** NTA showing **c** mode size and **d** particle concentration. The number of measured samples was 150, 100, 60, 60, 29, 60, and 30, respectively. The central line, box, errorbar, and dots indicate the median, inter-quartile range (Q1 and Q3), min–max range, and outliers, respectively. **e** SERS array. AuNPs were coated on the APTES-functionalized cover glass using centrifugation. Scale bar, 1 mm. **f** Representative SERS spectra. The center lines represent a mean of ten spectra from one sample to show general signal patterns. The light-colored area represents the standard deviation range.

throughput. To evaluate the SERS effect, signals of R6G solution were detected at conditions of SERS and spontaneous Raman (Supplementary Fig. 3a). The enhancement factor was calculated as $4.28 \times 10^5$. The uniformity in the signal acquisition was evaluated through the trend of 100 R6G signals scanned in the dot (Supplementary Fig. 3b). Since our substrate was fabricated based on colloidal nanoparticles, it has a non-uniform hotspot arrangement. Even in this non-uniform condition, a relatively uniform signal pattern was observed (Supplementary Fig. 3c). The scanned signals exhibited an identical tendency in intensity at the characteristic band (1364 cm$^{-1}$) of R6G, showing an averaged coefficient of variation (CV) of 6.0%. This uniformity may be due to the average signal enhancement of the nanoparticles in the laser-focused spot. In this circumstance, the number of nanoparticles is related to signal enhancement, and their average number of particles was $51 \pm 6$ particles/$\mu$m$^2$ in SEM characterization (Supplementary Fig. 3d).

To detect SERS signals of the isolated exosomes, exosomes were dropped onto each dot array, and their signals were scanned after thoroughly dried. Figure 2f shows representative spectra of exosomes isolated from each group. The common broad and strong signals near 860, 1283, and 1597 cm$^{-1}$ likely indicate citrate molecules on the AuNP surface or protein components such as tyrosine, phenylalanine, tryptophan, and amide III[37,38]. We observed subtle peak and intensity variations in several regions near 638, 668, 707, 733, 978, 1001, 1049, 1123, 1162, 1358, 1378, 1394, and 1432 cm$^{-1}$, which can be assigned to protein and lipid constituents[22,33,39,40]. Difference spectra were investigated to confirm the major difference in Raman signal between HC and cancer patients (Supplementary Fig. 4). As a result, we identified several Raman bands common across all cancer groups near 691, 826, 938,

961, 993, 1136–1152, 1245, 1527, and 1595 cm$^{-1}$. Most signal bands are assigned to protein constituents[41].

## Cancer presence diagnosis

Our approach is to recognize cancer patient samples by machine learning without specifying the characteristic band of the cancerous exosomes. The first step was to assess whether these data could be used to detect the presence or absence of cancer. As our approach is based on analyzing signals from random exosomes in plasma without selection, some signals may not reflect sample characteristics. In other words, some signals may be indistinguishable because of common exosomes derived from normal cells, even in cancer patient exosomes. Thus, we scanned 100 signals from one sample to capture the characteristic signal of cancerous exosomes as much as possible (see Fig. 1a). Then, by applying the multiple instance learning (MIL) concepts, an individual spectrum was collectively labeled with 0 for the control group and 1 for the patient group (Fig. 3a), then the average of predicted output derived from an individual sample was used as a single numerical value for diagnostic criteria. The neural network to implement the MIL was composed of a serial convolutional neural network (Supplementary Fig. 5a).

First, we investigated how many training samples are needed to predict unknown samples. For this purpose, the accuracy of independent samples was examined while increasing the number of training samples (Supplementary Fig. 5b). As a result, the accuracy tendency was saturated at over 30–40 samples per class. Based on these results, we set 50 for HC and 183 for cancer patients as the number of training samples to implement models. Accordingly, the entire sample was split into training ($n = 233$) and test ($n = 520$)

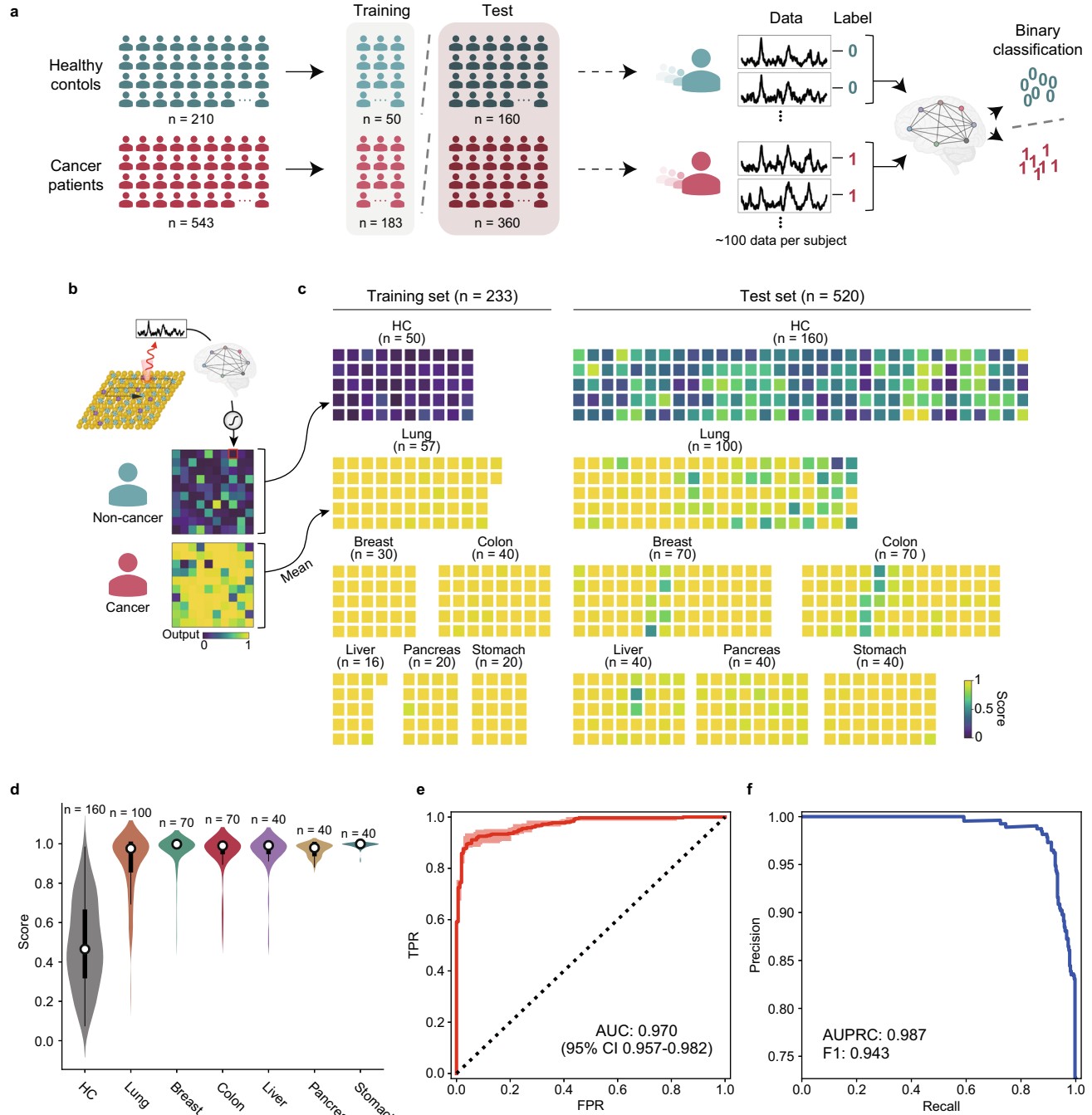

**Fig. 3 | Cancer presence detection. a** Training of the CNN classifier. Control and cancer patient samples were split into training ($n = 200$) and test ($n = 289$) sets. All data were comprehensively labeled by sample type. The CNN model was trained to produce a binary score of 0 or 1 using a sigmoid activation function. **b** Calculation of diagnostic scores. Model outputs corresponding to individual SERS signals are shown as a heatmap. Output means are used as final diagnostic scores. **c** Diagnostic scores for training (left) and independent test (right) samples. **d** Score tendency by cancer type. The inner box plot presents the median, inter-quartile range (Q1 and Q3), and min–max range. **e** ROC curve for the entire test set. The light-colored area represents a 95% confidence interval range. **f** PRC curve. Cartoons in panels **a** and **b** were created with BioRender.com.

samples (Fig. 3a). A total of 23,051 signals (4943 HC and 18,108 cancer samples) passed anomaly data filtering and were used for training.

Using this dataset, we trained the CNN-based binary classification model to detect the cancer presence. The data-wise training and validation accuracy reached over 90% (Supplementary Fig. 5c). The output of 100 scanned spectra per sample clearly shows the distinction between HC and cancer patients (Fig. 3b). Most of the signals had a value close to 0 in the HC sample, and vice versa in the cancer sample. The average score showed a distinct difference between HC and cancer

patients in both the training samples and the test samples (Fig. 3c). Cancer patient samples had 1.9-fold higher scores on average and showed a significant difference from HC samples, regardless of cancer type (Fig. 3d). The difference between sex in each group was not statistically significant (Supplementary Fig. 6).

Receiver operating characteristic (ROC) curves were calculated to evaluate diagnostic performance. For all HCs and cancer patients, the area under the curve (AUC) value was 0.970 [95% confidence interval (CI) 0.957–0.982] (Fig. 3e). Sensitivity and specificity at the optimal

cutoff were 89.4% and 96.3%, respectively. AUC values by cancer type were 0.936, 0.984, 0.972, 0.978, 0.992, and 0.999 for lung, breast, colorectal, liver, pancreatic, and stomach cancer, respectively (Table 1). The cutoff was determined based on the closest point to an ideal point in the ROC curve. The F1 score of the precision-recall curve (PRC) and the area under the PRC (AUPRC) value show that the model well predicts positive samples (Fig. 3f). Notably, the model retained effectiveness even under conditions of extreme specificity (Table 1). In 99% specificity condition minimizing false-positive diagnoses, the system had a sensitivity of 72.5%, suggesting that our approach enables the precise identification of multiple types of cancer.

For a better understanding of the behavior of the implemented model, we performed additional examinations of the implemented model. Firstly, to compare our model with other machine learning approaches, a dummy classifier and support vector machine (SVM) classifier were brought. In the prediction of the test dataset, our CNN-based model offers superior prediction compared to these two baseline classifiers (Supplementary Fig. 7). In addition, an ablation study was introduced to investigate the contribution of the layer to prediction; thus, we sequentially removed the layer parts of the model and monitored the performance (Supplementary Fig. 8a, b). Although overall convolution networks contributed to the performance, the first convolution layer was particularly influential (Supplementary Fig. 8b). Because the first layer is generally associated with basic feature detection of the input data and lessens unwanted features, the result indicates that this process is consistently important in Raman spectrum analysis as well[42,43]. In addition, we investigated the influence of structural parameters (Supplementary Fig. 8c). The filter size of the first layer is related to the window size skimming an input spectrum, and performance degrades above about a filter size of 10. The filter size of the other layer was saturated above 5. The dropout rate was consistent below 0.5, and the size of the FCL layer showed no significant effect on performance.

## TOO discrimination

Beyond classifying cancer presence, identifying cancer type is essential for precision diagnosis and appropriate treatment. Accordingly, we implemented an ensemble of models to classify cancer-identified samples by tissue of origin (Supplementary Fig. 9). Each model was built using the training dataset from the previous step; Subsequently, each model's diagnostic accuracy was calculated using the test dataset.

For the test samples, most cancer samples showed significantly higher scores for their corresponding tumor class (Fig. 4a). To evaluate model accuracy, ROC curves for one-vs-rest classification were used (Fig. 4b). Each model was able to predict its target class with an AUC value of 0.925, a sensitivity of 87.4%, and a specificity of 88.3%, on average (Table 2). Notably, the system had robust performance in identifying early-stage cancers (Fig. 4c). We measured classification accuracies for early-stage patients among the test samples based on clinical data, and early-stage samples included patients of TNM stage II or under. For liver cancer whose TNM information was missing, stage 0 (very early stage) and stage A (early stage) patients were selected using the BCLC staging system[44]. Notably, the models well predicted TOO even for early-stage cancer, reaching the mean AUC value of 0.945.

As with cancer presence detection, we compared the TOO discrimination performance against baseline classifiers. The dummy classifier poorly predicted the target class in all TOO discriminant cases, and the SVM classifier showed completely biased results in certain cancer types (breast and pancreatic cancer) as well. These results support that CNN-based classifiers have better performance not only in cancer presence but also in TOO discrimination.

Finally, we built a decision system to derive diagnostic predictions by integrating cancer presence detection and TOO discrimination. The decision rules for deriving predictions for individual samples are shown in Supplementary Fig. 11. First if the average of the cancer classifier predictions for an individual sample did not exceed the cutoff value, it was scored as non-cancer. Positive signals from samples exceeding the cutoff value were then input to the individual TOO classifiers. Since the models are not completely optimized, high prediction scores are output from two or more TOO models in single samples due to the prediction error. Therefore, final TOO predictions were determined based on the predicted class through a multi-layer perceptron (MLP) network to avoid false-negative diagnosis and guide to an appropriate precision diagnosis step. The MLP network was implemented based on the score value for TOO of the training sample, and the test samples for deriving the diagnostic accuracy were not used to implement the algorithm. The classification results derived from this decision rule for all test samples were summarized in a confusion matrix (Fig. 5). As a result, this system showed a sensitivity of 90.2% at a specificity of 94.4% while predicting the TOO of 72% of positive patients. In the analysis by cancer stage, the sensitivity for advanced-stage cancer patients was 97.5% (Supplementary Fig. 12). Notably, early-stage cancer patients were detectable with a sensitivity of 88.1% and a TOO accuracy of 75.9%, suggesting that our approach can be utilized as an early-stage diagnostic tool (Supplementary Fig. 12).

**Table 1 | Binary diagnostic performance for cancer presence for test samples**

| | # of samples | | AUROC | Sensitivity | Specificity | Accuracy | Precision | Sensitivity at 99% specificity | Specificity at 99% sensitivity |
|---|---|---|---|---|---|---|---|---|---|
| | − | + | | | | | | | |
| Total | 160 | 360 | 0.970 (0.957–0.982) | 0.894 (0.881–0.937) | 0.963 (0.964–0.968) | 0.915 (0.908–0.946) | 0.982 (0.981–0.986) | 0.725 (0.674–0.755) | 0.563 (0.180–0.611) |
| Lung cancer | | 100 | 0.936 (0.904–0.963) | 0.830 (0.765–0.915) | 0.919 (0.918–0.940) | 0.885 (0.858–0.931) | 0.865 (0.857–0.896) | 0.590 (0.500–0.617) | 0.563 (0.500–0.193) |
| Breast cancer | | 70 | 0.984 (0.966–0.997) | 0.957 (0.869–1.000) | 0.944 (0.994–0.950) | 0.948 (0.961–0.965) | 0.882 (0.981–0.896) | 0.857 (0.869–0.913) | 0.544 (0.473–0.950) |
| Colorectal cancer | | 70 | 0.972 (0.948–0.992) | 0.914 (0.887–0.964) | 0.975 (0.956–1.000) | 0.957 (0.935–0.987) | 0.941 (0.900–1.000) | 0.714 (0.690–0.964) | 0.575 (0.509–0.653) |
| Liver cancer | | 40 | 0.978 (0.948–0.997) | 0.950 (0.879–1.000) | 0.975 (0.970–0.988) | 0.970 (0.955–0.990) | 0.905 (0.853–0.952) | 0.675 (0.667–0.900) | 0.556 (0.497–0.988) |
| Pancreatic cancer | | 40 | 0.992 (0.980–1.000) | 1.000 (1.000–1.000) | 0.969 (0.948–0.988) | 0.975 (0.960–0.990) | 0.889 (0.855–0.950) | 0.650 (0.447–0.974) | 0.969 (0.948–0.988) |
| Stomach cancer | | 40 | 0.999 (0.995–1.000) | 1.000 (1.000–1.000) | 0.981 (0.969–1.000) | 0.985 (0.975–1.000) | 0.930 (0.886–1.000) | 0.975 (0.821–1.000) | 0.981 (0.969–1.000) |
| (95% CI) | | | | | | | | | |

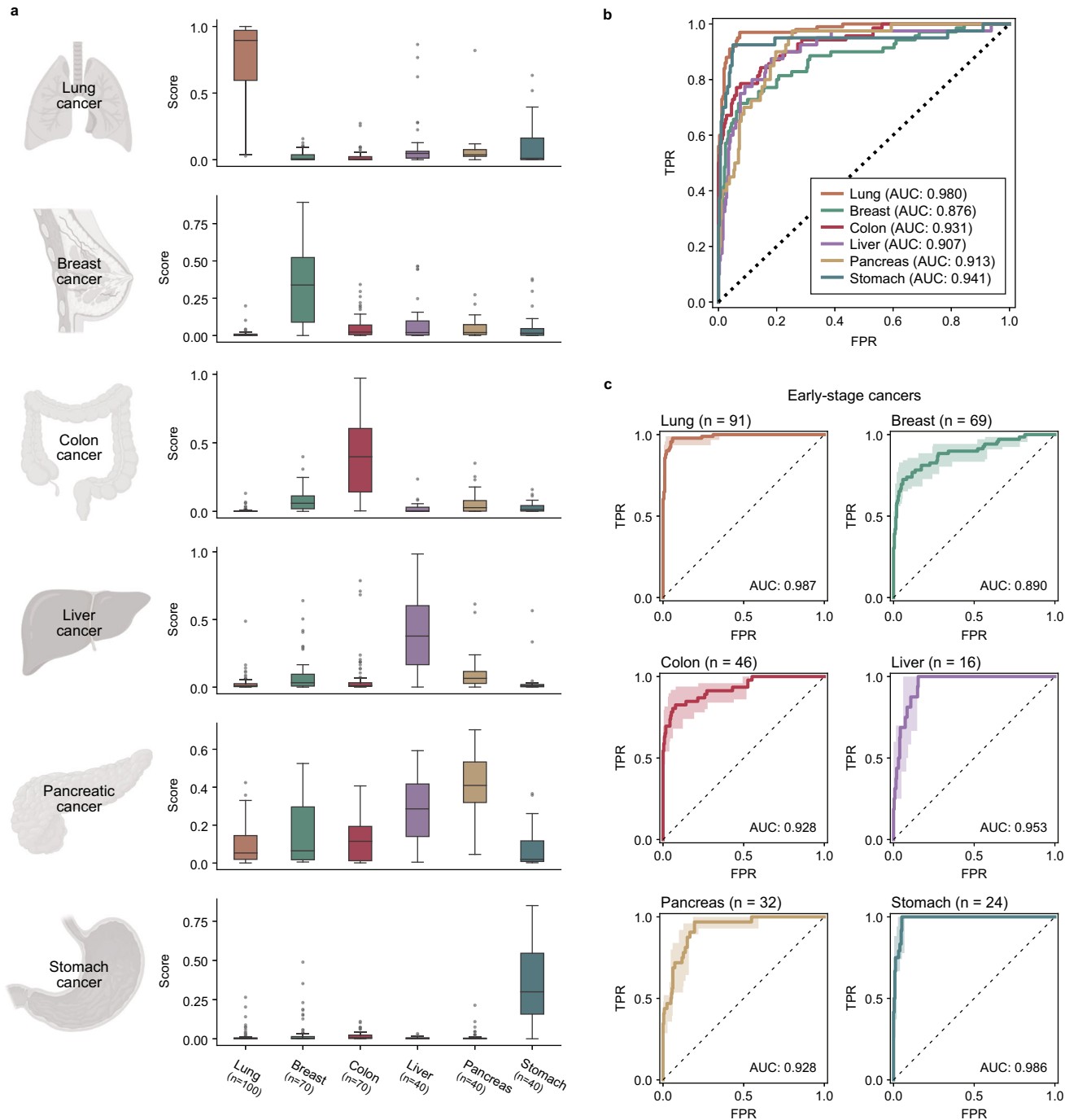

**Fig. 4 | Tissue of Origin (TOO) discrimination. a** Diagnostic score tendency for each cancer type for test samples. The box plot presents the median, inter-quartile range (Q1 and Q3), min–max range, and outliers. Bars represent standard errors. **b** ROC curves showing classification performance for the entire test sample. AUCs are shown in each graph. **c** Classification of early-cancer cases. Lung, breast, colon, pancreatic, and stomach cancer cases involve stages 0–II using the TNM staging system based on the AJCC staging system. Liver cancer cases include stages 0 and A, according to the Barcelona clinic liver cancer (BCLC) staging system[44]. The light-colored area represents a 95% confidence interval range. Cartoons in panel (**a**) were created with BioRender.com.

## Discussion

Recently, methods for detecting circulating biomarkers (e.g., ctDNA and CTC) based on whole-genome sequencing or DNA methylation patterns are emerging for liquid biopsy[45,46]. Despite the innovation in detection methods, accurate diagnosis of early-stage cancer remains challenging. Exosomes are attractive because they are relatively free from problems associated with other biomarkers, such as low abundance and undesired fragmentation, based on their properties of being actively secreted from living tumor cells, even at early stages, and having cargoes protected by a lipid bilayer[47,48]. In addition,

exosome stability in long-term storage or freezing of clinical samples provides another advantage for their clinical application[47,49].

Meanwhile, SERS has been applied for cancer diagnosis and liquid biopsy in various ways. Many groups have tried to detect cancerous biomolecules in blood, urine, saliva, and breath through SERS for medical diagnostic application[50]. Recently, based on the aforementioned advantages of exosomes as a biomarker, reports on the SERS characterization for cancerous exosomes are increasing[20]. In this stream, several papers have been reported on early-stage detection, including breast cancer[51,52], colon cancer[53], and lung cancer[33] based on

**Table 2 | TOO performance for test samples**

| | # of samples | | AUROC | Sensitivity | Specificity |
|---|---|---|---|---|---|
| | − | + | | | |
| Lung cancer | 260 | 100 | 0.980 (0.964–0.992) | 0.970 (0.899–0.938) | 0.927 (0.952–0.992) |
| Breast cancer | 290 | 70 | 0.876 (0.814–0.924) | 0.714 (0.766–0.826) | 0.931 (0.780–0.924) |
| Colorectal cancer | 290 | 70 | 0.931 (0.896–0.961) | 0.786 (0.712–0.838) | 0.924 (0.922–0.962) |
| Liver cancer | 320 | 40 | 0.907 (0.847–0.955) | 0.875 (0.781–0.932) | 0.819 (0.921–0.886) |
| Pancreatic cancer | 320 | 40 | 0.913 (0.867–0.949) | 0.975 (0.980–0.970) | 0.747 (0.712–0.869) |
| Stomach cancer | 320 | 40 | 0.941 (0.880–0.988) | 0.925 (0.821–1.000) | 0.950 (0.944–0.957) |
| (95% CI) | | | | | |

the SERS profiles of the bio-liquid itself or its components like exosomes. However, issues of localization of cancer types in the early stage and a small number of test samples remain. We attained effective discrimination between 6 early-stage cancer types and diagnostic sensitivity and specificity of over 90% with 520 test samples not used to train algorithms. This diagnostic system offers clinicians the opportunity to select tissues that need more detailed examination before the advanced stage of tumors without the time and expense of multiple tests.

As a diagnostic method, our approach provides additional advantages. First, it is rapid. Because we utilized an automated system, exosome isolation (20 min) and detection (30 min for specimen preparation; 10 min for detection) can be completed in an hour. The final decision can be completed in several seconds using pre-trained AI models. The SERS chip with 10 measurement spots is not reusable, but it would be possible for automated detection and diagnosis through programmed stage control. Second, it reduces the resources required for diagnosis. Our label-free SERS requires no additional extra reagents for amplifying or capturing analytes; it only requires a concise experimental procedure of dropping 10 µL of exosome solution on a small hot-spot array. Therefore, the biochemical resources required for diagnosis can be greatly reduced, in turn reducing cost. Third, expansion to more diverse diseases is likely possible, as it is not difficult to insert an additional diagnostic module for identifying other diseases from the SERS signal. This offers the advantage of expanding the diagnostic scope to other diseases that may induce disturbance of the nature and subpopulation of exosomes in plasma.

Nevertheless, several limitations remain to be addressed: first, there is still a need to augment a larger number of training samples, and diagnostic performance must be verified using external tests and prospective clinical trials; second, potential confounding factors, including those generated by benign tumors, should be considered; third, major spectral features should be examined, and its association with reported biological pathways or factors from the pathology perspective; fourth, the introduction of uniform and mass-manufacturable SERS detection chips may be required for precision and reproducibility in clinical practice. Ultimately, it will be essential to establish a well-controlled diagnostic process using the actual clinical workflow from blood draw to diagnostic report.

In summary, we have demonstrated the simultaneous diagnosis of six types of early-stage cancers using a single spectroscopic detection. We implemented and evaluated an AI diagnostic system for hundreds of human plasma samples. We achieved a high diagnostic accuracy of over 95% and TOO discrimination performance. Importantly, the system was effective even for early-stage cancer patients. We hope that this method can provide appropriate precision diagnoses to more potential patients and improve their prognoses.

## Methods
### Human plasma samples
The Institutional Review Board of Korea University Guro Hospital approved this study (approvals 2020GR0176 and 2021GR0013 for healthy participants and cancer patients, respectively). We obtained written informed consent from all participants who underwent blood collection. This study used a total of 210 HC and 543 cancer patient blood plasma samples (Table S1).

Inclusion criteria of this research include (1) an adult of Korean nationality, (2) patients who received cancer surgery and permanent pathology of lung cancer (adenocarcinoma), breast cancer (duct carcinoma), colon cancer (adenocarcinoma), liver cancer (hepatocellular carcinoma), pancreatic cancer (duct carcinoma), and stomach cancer (adenocarcinoma). (3) Patients without neoadjuvant therapy before the cancer surgery, and (4) patients who have not been diagnosed with other cancers before cancer surgery. Exclusion criteria are patients who do not meet the inclusion criteria. The sex of participants was defined based on self-report.

The eligible biospecimen of cancer patients were retrospectively and randomly obtained according to inclusion/exclusion criteria through 3 human biobanks (Biobank of Korea University Guro Hospital, Asan Bio-Resource Center, and Biobank of Ajou University Hospital) in the Republic of Korea. Blood plasma samples from HCs without a personal cancer history were retrospectively randomly obtained from the Korea Institute of Radiological and Medical Sciences (KIRAMS) Radiation Biobank and the Biobank of Seoul National University Bundang Hospital, Republic of Korea. The plasma samples were collected before surgery after a permanent pathology was confirmed. Since this study used retrospectively collected samples, the period of time for recruitment and data collection was not established. Since this study was a pilot study to develop a method to identify multiple cancers is possible, no sample size calculation to explore clinical utility was performed. All samples were stored at −80 °C.

To implement models, clinical information was available to train models with actual labels. The predictions on the test sample were made without information on the correct class.

### Exosome isolation
Frozen suspensions were thawed at 4 °C. The exosome isolation was performed using a size-exclusion chromatography column (Exo-I S5, Exopert, KR)[36]. After the substitution of the inner liquid, 500-µL of plasma was loaded onto the prepared column. When the plasma was permeated into the column thoroughly, PBS was added as a mobile phase. Then, 500-µL of the eluted fractions were collected serially. Fractions as the exosome suspension were selected after the evaluation of collected particles. The resulting suspensions were stored at −80 °C until subsequent analysis.

### Exosome evaluation
Western blotting of the isolated exosome. Isolated exosomes were lysed in DBPS without calcium chloride and magnesium chloride (WELGENE, South Korea). Proteins were determined using Bradford Dye Reagent (Bio-Rad, USA) and boiled with 5× SDS loading buffer (CELLNEST, South Korea) for 5 min at 95 °C. Totally, 20 µg of exosomal total proteins were separated in 4–20% precast protein gel (Bio-rad, USA) and transferred onto PVDF membranes (Bio-rad, USA). The

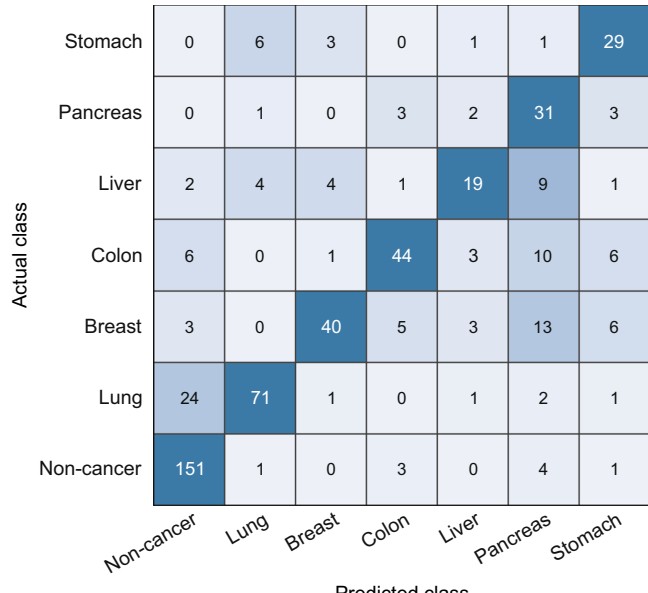

**Fig. 5 | Confusion matrix for the entire test sample.** Prediction results for 520 untrained test samples through the final decision model combining cancer presence detection and TOO discrimination. Cancer presence was determined by whether the mean score was greater than the cutoff value. TOO was determined based on a multiclassification result built through a multi-layer perceptron network.

membranes were blocked using 3% BSA in TBS containing 0.1% Tween 20 for 1 h at room temperature. Anti-CD9, CD63, CD81, TSG101, and calnexin antibodies were used at a dilution of 1:1000. Anti-ApoA, ApoB, and HSA antibodies were used at a dilution of 1:3000. The membranes incubated with the diluted primary antibodies in 2% BSA solution for overnight at 4 °C. The membranes were then incubated with HRP-labeled secondary antibodies, goat anti-mouse IgG for exosomal proteins and HSA, and goat anti-rabbit IgG for lipoproteins (CELLNEST, South Korea) at dilutions of 1:10,000 and developed by chemiluminescence substrate (Bio-Rad, USA). The signals were captured using the bio-imaging system(Amersham ImageQuant 800, cytiva, Germany).

For TEM, exosome suspensions were fixed using 0.1% (v/v) paraformaldehyde at a 1:1 volume ratio for 30 min. A 200 mesh Lacey carbon grid was immersed in 10 μL of fixed exosomes for 7 min. The residual suspension was soaked through gentle touching with filter paper and thoroughly rinsed with deionized water. The resulting grid was then immersed in 10 μL of 1% UranyLess EM stain solution for 7 min. The residual solution was soaked with filter paper and dried. TEM was performed with an FEI Tecnai F20 G2.

For NTA, the Nanosight NS300 instrument (Malvern Panalytical Ltd.) was used with samples diluted 100–1000-fold.

### SERS detection chips

Cover glasses were cleaned by immersion in piranha solution ($H_2SO_4$:$H_2O_2$ = 3:1) for 30 min to eliminate organic impurities, then thoroughly rinsed with deionized water and ethanol. The cover glasses were then immersed in 1%(v/v) 3-Aminopropyltriethoxysilan (APTES) ethanoic solution to functionalize the surface, then thoroughly rinsed with ethanol and dried using $N_2$ gas. Polydimethylsiloxane (PDMS) wells were prepared by punching holes at regular intervals with a 2.5 mm diameter biopsy punch. The resulting PDMS wells were attached to the APTES-functionalized substrate. Next, 100 nm AuNP colloidal solution (NanoComposix) was concentrated 5-fold through centrifugation. Then, 8 μL of the concentrate was added to each hole array. The good substrate with AuNP solution was placed on a

centrifuge rotor configured to hold the substrate and then centrifuged at 1000×*g* for 5 min. After the PDMS well was detached, the substrate was rinsed with deionized water and dried. Exosome suspension was dropped onto each dot and dried at 35 °C.

### SERS

The Raman signal was observed with Axio Oberver3 equipped with iDus420 CCD (Andor iDus420) and spectrograph monochromator Monora322i (Dongwoo Optron). Laser irradiation was performed at 2 mW and 785 nm through a 50× objective lens (NA 0.7). The acquisition time was 1 s per single take. To maintain focus and automated signal scanning, customized software was built using Python and the pyQT5 library. Signals were preprocessed for denoising, baseline correction, and elimination of spiked data. Anomalous data were excluded if the intensity of the common band near 860 cm$^{-1}$ did not exceed a threshold specified manually through signal comparison.

### Neural network algorithm

All data were handled through custom Python code. The neural network model, including CNN, SVM, MLP, and dummy classifier, were implemented using Python library, including scikit-learn, TensorFlow 2.5, and Keras API. The model architecture was composed of serial convolutional layers to conduct binary classification through a sigmoid activation function. 20% of the training data was used as a validation dataset to monitor overfitting during the learning process. All hyperparameters, including learning rate, decay rate, and training epoch, were optimized through manual and random searches.

### Statistics and reproducibility

Multivariate and statistical analyses (e.g., ROC curve, AUC calculation, PRC) were performed using a scikit-learn library in Python. The statistical analysis, including the *t*-test, was performed using Python scipy and pingouin library. All visualizations of data were made using Python and MATLAB R2021a (MathWorks) codes. The western blotting data and TEM images in Fig. 2 were collected from at least two independent experiments.

### Reporting summary

Further information on research design is available in the Nature Portfolio Reporting Summary linked to this article.

## Data availability

The datasets generated during and analyzed during the current study are available in the GitHub repository. Source data are provided in this paper.

## Code availability

All the code used for prediction using implemented algorithms and generation of figure data is available from the GitHub repository with sample data for the demo.

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

## Acknowledgements

This study was supported by a grant from the Seoul R&BD Program through the Seoul Business Agency (SBA) funded by the Seoul Metropolitan Government (BT210040, PI: H.S.) and the Korea Medical Device Development Fund grant funded by the Korean government (the Ministry of Science and ICT, the Ministry of Trade, Industry and Energy, the Ministry of Health & Welfare, the Ministry of Food and Drug Safety) (Project Number: 1711174279, RS-2020-KD000094, PI: Y.C.). The biospecimen and data used in this study were provided by the Korea Institute of Radiological and Medical Sciences (KIRAMS) Radiation Biobank (KRB, KRB-2021-E002), the Human Biobank of Seoul National University Bundang Hospital (Distribution No. DT-2020-013-01), the Biobank of Korea University Guro Hospital, the Asan Bio-Resource Center (2021-02(219)), and the Biobank of Ajou University Hospital, a member of Korea Biobank Network.

## Author contributions

H.K.K. and Y.C. designed the study. H.S. developed the concept. B.H.C. collected clinical samples and data. H.S. and B.H.C. wrote the paper. O.S. obtained the SERS dataset. H.S. and O.S. wrote code and implemented the AI models. H.S. analyzed output data. J. K. performed biological experiments and evaluations. H.S. conducted data visualization and made the figures. H.K.K., S.K.C., and Y. P. gave clinical advice. S.K.C. collected healthy control samples. H.K. and Y.C. conducted a funding acquisition. H.K.K. and Y.C. conducted project administration. H.S. and B.H.C. contributed equally.

## Competing interests

Yeonho Choi, Hyun Koo Kim, and Yong Park hold equity in EXoPERT Corporation. The remaining authors declare no competing interests.
