## [Peer Review File · Nature Communications]

Single test-based early diagnosis of multiple cancer types
using Exosome-SERS-AIREVIEWER COMMENTS

Reviewer #1 (Remarks to the Author):

The early cancer detection from biopsy is an important and trendy topic in recent years. In particular, there are also different commercialization for it in recent years. The authors have proposed an approach to detect early cancers and its cancer types from liquid biopsy based on exosome with SERS. The study has most of the necessary components. The main novel/competitive features here lies in two aspects. (1) Early cancer detection and localization based on SERS (2) Very strong detection performance > 90%. Overall, I can see that the authors have demonstrated their efforts with due respects. The followings are my suggestions:

1. For the SERS profiles, have other scientists adopted those SERS profiles for early cancer detection before? If yes, the feature (1) novelty could be diminished and should be critically discussed and compared.
2. For the feature (2), the detection performance has to be verified further in different machine learning settings. In particular, the major concern here lies in the sample size. Since it covers 6 cancer types, if we have 289 test samples, it literally means we may have less than 50 test samples for each cancer type in effect. Its generality has to be carefully support such as the addition of extra test samples, or sample bootstrapping, or other methods.
3. The proposed neural network architecture has been visualized in the supplementary. However, to support the feature (2), the authors may wish to perform the corresponding parameter analysis and ablation analysis to ascertain that the network architecture is effective. Other baseline machine learning methods should also be benchmarked and compared.
4. To be compatible with Nature Communications, a high degree of scientific rigors and reproducibility is expected. Therefore, source code and data should be opened on GitHub / GitLab / CodeOcean.
5. Right now, the proposed approach looks like a black-box. It is also important to uncover how the proposed approach made decisions from the pathology perspective. Feature importance analysis with WikiPathways may help on deciphering what the proposed approach has learned from SERS, in relation to the AUROCs. On the other hand, the AUPRCs of PRC curves should also be provided.

Reviewer #2 (Remarks to the Author):

The work of Shin et al. is very interesting and well-designed. In the last years, extracellular vesicles, with emphasis on exosomes (EXOs), are arousing huge interest in cancer diagnosis and vibrational spectroscopies, such as Raman spectroscopy, may be a promising tool for a label-free characterization of these vesicles.

In this work, the Authors exploit surface-enhanced Raman spectroscopy (SERS) for the spectral profiling of EXOs. EXOs were isolated from the serum of a large cohort of healthy donors and cancer patients affected with different types of cancer.

Efficient isolation, and characterization, of EXOs, is a crucial step in works focused on extracellular

vesicles. In this case, the Authors used size exclusion chromatography for the EXO isolation, without chemical treatments and chance to introduce artefacts in the samples, as it may happen by using other isolation techniques, such as precipitation kits. In accordance with MISEV 2018 guidelines (<https://doi.org/10.1080/20013078.2018.1535750>), the Authors performed an appropriate characterization of EXOs by western blotting, TEM and NTA.

SERS signals from EXOs were analysed by a machine learning approach providing high diagnostic accuracy and tissue of origin discrimination performance.

This work may pave the way for the development of novel diagnostic techniques by a comprehensive and label-free EXO characterization. Then, I strongly support the publication of this work.

I have just some minor revisions to suggest:

- 1) The authors should highlight the improvement of their work with respect to the current literature. Are they the first to use Raman spectroscopy for EXO spectral profiling in a such large cohort of cancer patients? Have been differences between the Raman spectra of cancer and non-cancer EXOs (e.g., EXOs from cancer and normal cells) already demonstrated? Please refer to <https://doi.org/10.3390/nano11061476>.
- 2) The use of artificial intelligence for the development of EXO-based diagnostic techniques is extremely interesting and promising. The Authors should check if some other works combined vibrational spectroscopies, Raman or FTIR, and machine learning-assisted analysis of circulating extracellular vesicles for cancer diagnosis.
- 3) Check the numbering of supplementary figures. Supplementary figure 8 should be Supplementary figure 1, Supplementary figure 9 should be Supplementary figure 2, and so on.
- 4) Samples of EXOs isolated from serum include not only cancer-derived EXOs but vesicles released by blood cells. Have the authors considered the presence of non-cancer EXOs and how they possibly affect the analysis?
- 5) In line 182 the Authors state that “the system had robust performance in identifying early-stage cancers (Fig. 4c)”. Despite the sample size of the cancer patients being very high, in this case, the number of liver and stomach cancer patients at early-stage is very low (3 and 7 respectively). Please take this into account.

Reviewer #3 (Remarks to the Author):

The presented manuscript deals with a very promising and interesting field, that is liquid biopsy. The results are well presented and in general the proposed approach sounds technically interesting and

biologically challenging. Indeed, exosomes are very interesting cellular vesicles supposed to be potential important actors in several biological pathways. Above all their capability to travel in the plasma give them a very important role in cellular signalling at large distances.

The application of AI algorithms is quite convincing and looks a robust method for analysing spectral data, in order to generate/identify specific disease (cancer) patterns.

The use of SERS technique could seem of crucial importance for achieving a reasonable signal level from exosomes samples, but this reviewer has several concerns about the accomplishment of SERS detection. Hence, there is enough significance in the achieved results, but some major revisions are required.

MAJOR REVISIONS:

1) Lines 110-112 and Figure2f:

The representative spectra look much more like standard Raman spectra than enhanced spectra. Enhanced spectra by SERS are usually characterized by sharper peaks, or at least by some sharper peaks. Despite of the average process, 10 items for the average should leave some sharper features observable in the spectra (due to higher SERS intensities). Usually, SERS spectra require quite low laser powers, in the 0.1-0.5mW range. The laser power of 2mW (reported in materials and methods) is not very high, but also not so low for exciting truly SERS signal. How the author could be sure that the SERS substrate is properly working and providing enhanced spectra, instead of conventional, non-enhanced spectra? and what about the enhancing factors? are they evenly distributed on the substrates or are they rather non-uniform?

2) Lines 164-166 and Supplementary Figure6:

The authors claim that: "We found that true-positive signals tended to deviate from the normal cluster of true-negative signals and were located on the positive side of the principal component 2 (PC2) axis." This is only partly true. From the figure one can say rather the opposite: that true positive signals are covering the whole range (positive and negative PC2 scores as well as PC1 positive and negative scores), while the true negative signals (HC samples) are only in the negative PC2 range. But there is a large area where HC sample and Cancer samples are overlapping. The authors should re-arrange the statements/results of PCA in less resolute sentences since these results are quite weak from a spectral point of view.

3) Line 261:

This reviewer did not find any info online about the reported exosome isolation kit "Exo-I S5 kit (Exopert)": should be provided a direct link to this product? how is working this kit? are precipitation, isolation, buffer solutions involved? have been they tested for Raman spectroscopy? It is worthy to point out that every possible source of interference with the spectroscopic results should be addressed and mentioned in the manuscript.

4) Lines 278-279:

In the Materials and Methods section, the authors mention the Au NP size: "100 nm AuNP colloidal solution (NanoComposix) was concentrated 5-fold through centrifugation..." 100nm is quite a large size to achieve SERS effect directly from the single Au-NPs, so it is reasonable that

the gaps between NPs (expected to be in the 10nm range) could be responsible of Raman enhancements. But in this case the NPs layer geometry plays a crucial role. Do the author have an idea of NPs arrangement? in the inset of fig. 2e, the SEM image of NPs shows a random arrangement that could lead to different enhancing factors randomly distributed over the SERS chip. Could the author comment a little bit more on the enhancing properties of the chip? and about their uniformity over the chip area?

The figure "SERS substrate signal uniformity" in the Supplementary shows a quite uniform signal from rhodamine6g, but again the spectra do not own the specific features of SERS signals, but rather they look like standard Raman spectra of rhodamine6g. And again, there is the doubt that the acquired spectra are effectively enhanced Raman signals.

5) Lines 286-287:

As already mentioned above, the reported laser power is slightly too high for achieving genuine SERS signals rather than simply conventional Raman spectra. Did the author consider the case that they are recording conventional Raman signals? do they have recorded some comparison spectra (without SERS) to show the effect of the enhancement?

MINOR REVISIONS:

6) In the Conclusion section: Could the authors comment on the reusability of the device? Is it reusable after proper washing, or is it intended for single use (disposable device)?

7) Line 131: MIL should be defined as acronym, is it Multiple Instance Learning?

Reviewer #4 (Remarks to the Author):

The article reports on the use of SERS and AI to classify cancer tissue. The workflow is good and the results are convincing. I believe this makes a strong contribution to the field. I only have minor concerns to be addressed.

I think one of the limitation is the number of samples used for training and for the validation. While I understand that the number of spectra is relatively standard, the number of clinical samples in the case of some cancer is low. How the model holds on with larger number of samples remains to be shown and this need to be highlighted in the limitation section.

Related to this, it is not clear how the repeat spectra were collected from samples to reach nearly 20,000 signals. Is it from multiple spectra on a single spot or from multiple samples?

The ratio of training to test is unusual, where fewer training samples were used. Could the author explain why?

In Fig 2, it would be nice to have the differential spectra from each cancer type in relation to HC. The differences mentioned in lines 111 to 114 would be more evident.

I don't agree with the statement that PCA identifies the signals used by the ML to classify. Both are different mathematical models and there is a strong chance that different weights are given to different Raman frequencies in the classification. This needs to be better supported or corrected.

Lines 182 to 187, I am worried that the sample number is too low to make such discrimination.

Figure numbering in SI is incorrect. It starts at S8 and there are two Figures S8

Page numbers are striked in SI.

Response to comments

I. Reviewer #1's comments

I-0

The early cancer detection from biopsy is an important and trendy topic in recent years. In particular, there are also different commercialization for it in recent years. The authors have proposed an approach to detect early cancers and its cancer types from liquid biopsy based on exosome with SERS. The study has most of the necessary components. The main novel/competitive features here lies in two aspects. (1) Early cancer detection and localization based on SERS (2) Very strong detection performance > 90%. Overall, I can see that the authors have demonstrated their efforts with due respects. The followings are my suggestions:

- ***Response:*** We appreciate the time and effort for the review. The reviewer's comments provided useful insight to strengthen our paper. Considering the comments, we revised the manuscript and several figures and added point-by-point answers.

For the SERS profiles, have other scientists adopted those SERS profiles for early cancer detection before? If yes, the feature (1) novelty could be diminished and should be critically discussed and compared.

- **Response:** Thank you for the comment. For cancer diagnosis and liquid biopsy, SERS has been applied in various ways. Many groups have tried to derive the meaningful signal from biomolecules in many bio-liquid (blood, urine, ...). Also, SERS is emerging technique to detect exosomes for liquid biopsy. In this stream, several papers have been reported on early-stage cancer detection (Analytical and bioanalytical chemistry 407.24 (2015): 7503-7509.; ACS applied materials & interfaces 12.7 (2020): 7897-7904.; Journal of Biophotonics 13.4 (2020): e201960176.). We also reported a paper about early-stage lung cancer diagnosis (ACS nano 14.5 (2020): 5435-5444.). Although these papers show high potential for interpreting SERS profiles for biomolecules, several issues of [1] localization of cancer types in the early-stage and [2] a small number of test samples (N: < 100) remain. To overcome the issues, we attained [1] effective discrimination between 6 early-stage cancer types, and [2] high diagnostic accuracy with 520 test samples not used to train algorithms. Agreeing with the reviewer we thought it worthy of mentioning this point in the discussion section.

Original manuscript ⇒ Revised manuscript

“Our diagnostic system offers clinicians the opportunity to select tissues that need more detailed examination without the time and expense of multiple tests. Recently, methods for detecting circulating biomarkers (e.g., ctDNA, CTC) in liquid biopsies, based on whole-genome sequencing or DNA methylation patterns, are emerging for this purpose^{39,40}. Despite the innovation in detection methods, accurate diagnosis of early-stage cancer remains challenging. Exosomes are attractive because they are relatively free from problems associated with other biomarkers, such as low abundance and undesired fragmentation, based on their properties of being actively secreted from living tumor cells, even at early stages, and having cargoes protected by a lipid bilayer^{41, 42}. In addition, exosome stability in long-term storage or freezing of clinical samples provides another advantage for their clinical application^{41, 43}. Based on these advantages, we implemented a viable diagnostic system based on exosome-SERS-AI and demonstrated its effectiveness for liquid biopsies for multiple types of early-stage cancers.” – p. 14 Lines 212-223

⇒ *“Recently, methods for detecting circulating biomarkers (e.g., ctDNA, CTC), based on whole-genome sequencing or DNA methylation patterns, are emerging for liquid biopsy^{45, 46}. Despite the innovation in detection methods, accurate diagnosis of early-stage cancer remains challenging. Exosomes are attractive because they are relatively free from problems associated with other biomarkers, such as low abundance and undesired fragmentation, based on their properties of being actively secreted from living tumor cells, even at early stages, and having cargoes protected by a lipid bilayer^{47, 48}. In addition, exosome stability in long-term storage or freezing of clinical samples provides another advantage for their clinical application^{47, 49}.*

Meanwhile, SERS has been applied for cancer diagnosis and liquid biopsy in various ways. Many groups have tried to detect cancerous biomolecules in blood, urine, saliva, and breath through SERS for medical diagnostic application⁵⁰. Recently, based on the aforementioned advantages of exosomes as a biomarker, reports on the SERS characterization for cancerous exosome are increasing²⁰. In this stream, several papers have been reported on early-stage

detection, including breast cancer^{51, 52}, colon cancer⁵³, and lung cancer³³ based on the SERS profiles of the bio-liquid itself or its components like exosomes. However, issues of localization of cancer types in the early-stage and a small number of test samples remain. We attained effective discrimination between 6 early-stage cancer types, and high diagnostic accuracy with 520 test samples not used to train algorithms. This diagnostic system offers clinicians the opportunity to select tissues that need more detailed examination before the advanced stage of tumors without the time and expense of multiple tests.” – p. 17 Lines 267-286

For the feature (2), the detection performance has to be verified further in different machine learning settings. In particular, the major concern here lies in the sample size. Since it covers 6 cancer types, if we have 289 test samples, it literally means we may have less than 50 test samples for each cancer type in effect. Its generality has to be carefully support such as the addition of extra test samples, or sample bootstrapping, or other methods.

- ***Response:*** We agree with the reviewer's opinion. Previously, we did not have enough test samples to verify our approach and support early-stage cancer detection. Considering these concerns, we greatly increased the test sample from 289 to 520 through additional sample collection and signal detection. In particular, the number of samples of colon, liver, and stomach cancer, which had a significantly small number of initial samples, was increased several times. (N: Colon 16 → 46, Liver 3 → 16, Stomach 7 → 24) The resulting AUC of ROC was 0.97, and the presence of 6 cancers could be detected with high accuracy even in a larger number of samples. In addition, it was confirmed that tissue of origin detection was possible with an average AUC of above 90%.

Original manuscript ⇒ Revised manuscript

“Here, we demonstrate a liquid biopsy method that combines AI and SERS to simultaneously diagnose multiple cancer types by label-free analysis of plasma exosomes (Fig. 1a). Our method acquires SERS signals of isolated exosomes, then analyzes them with multiple instance learning (MIL)-based deep learning models. There are two outputs: cancer diagnosis and tissue of origin (TOO) discrimination. (Fig. 1b) In the first step, the deep learning model classifies each signal as normal or cancerous, yielding a score of cancer presence. In the second step, multiple classifier models trained cancer types using the one-vs-rest method generate TOO determinations of positive predictions from the first step. In this paper, we demonstrate the diagnostic performance of this system using 289 test samples that had not been used for training. The samples include 6 cancer types (lung, breast, colon, liver, pancreas, and stomach) and early-stage cancer patients.”

– p. 4 Lines 61-70

- ⇒ *“Here, we demonstrate a liquid biopsy method that combines AI and SERS to simultaneously diagnose multiple cancer types by label-free analysis of plasma exosomes (Fig. 1a). Our method acquires SERS signals of isolated exosomes, then analyzes them with deep learning models. There are two outputs: cancer diagnosis and tissue of origin (TOO) discrimination. (Fig. 1b) In the first step, the deep learning model classifies each signal as normal or cancerous, yielding a score of cancer presence. In the second step, multiple classifier models trained cancer types using the one-vs-rest method generate TOO determinations of positive predictions from the first step. In this paper, we demonstrate the diagnostic performance of this system using 520 test samples that had not been used for training. The samples include 6 cancer types (lung, breast, colon, liver, pancreas, and stomach) and early-stage cancer patients.”* – p. 4 Lines 65-74

“Fig. 3: Cancer presence diagnosis. *a*, Training of the CNN classifier. Control and cancer patient samples were split into training ($n = 200$) and test ($n = 289$) sets. All data were comprehensively labeled by sample type. The CNN model was trained to produce a binary score of 0 or 1 using a sigmoid activation function. *b*, Calculation of diagnostic scores. Model outputs corresponding to individual SERS signals are shown as a heatmap. Output means are used as final diagnostic scores. *c*, Diagnostic scores for training (left) and independent test (right) samples....” – p. 9

“Fig. 3: Cancer presence detection. a, Training of the CNN classifier. Control and cancer patient samples were split into training (n = 233) and test (n = 520) sets. All data were comprehensively labeled by sample type. The CNN model was trained to produce a binary score of 0 or 1 using a sigmoid activation function. b, Calculation of diagnostic scores. Model outputs corresponding to individual SERS signals are shown as a heatmap. Output means are used as final diagnostic scores. c, Diagnostic scores for training (left) and independent test (right) samples.” – p. 12

“Fig. 4: Tissue of origin (TOO) discrimination. ... c, Classification of early-cancer cases. Lung, breast, colon, pancreatic, and stomach cancer cases involve stages 0, I, and II using the TNM staging system based on the AJCC staging system. Liver cancer cases include stage A, according to the Barcelona clinic liver cancer (BCLC) staging system” – p. 12

“Fig. 4: Tissue of origin (TOO) discrimination. ... c, Classification of early-cancer cases. Lung, breast, colon, pancreatic, and stomach cancer cases involve stages 0, I, and II using the TNM staging system based on the AJCC staging system. Liver cancer cases include stage 0 and A, according to the Barcelona clinic liver cancer (BCLC) staging system⁴⁴.” – p. 15

The proposed neural network architecture has been visualized in the supplementary. However, to support the feature (2), the authors may wish to perform the corresponding parameter analysis and ablation analysis to ascertain that the network architecture is effective. Other baseline machine learning methods should also be benchmarked and compared.

- **Response:** Thanks for the reviewer comments. As recommended by the reviewer, we performed parameter analysis to investigate the effects of kernel size, dropout rate, and fully connected layer size and ablation analysis through artificial omission of layers for a deeper understanding of how the neural network works. As a result, we were able to add a complementary explanation of the factors influencing the accuracy trend in the neural network. We have added these new result and description to the result section (p. 10-11 Lines 185-200).

Additionally, to rigorously compare the performance of these CNN-based neural networks per the reviewer's recommendation, we brought in a dummy classifier and a support vector machine (SVM) as baseline machine learning methods. As a result, these baseline models had a problem of providing biased results, so it was confirmed that our CNN model have superior performance in the plasma exosome SERS signal analysis. We have added these results to the main text and supplementary information. (p. 10-11 Lines 185-200; p. 13 Lines 226-230).

Added manuscript

“Supplementary Fig. 7: Comparison with other models in cancer detection. As a baseline model, dummy classifier and support vector machine (SVM) were established. The accuracy represents data-wise prediction accuracy for the test dataset in each model.”

- SI p. 10

“Supplementary Fig. 8: Analysis for architecture and performance. (a) Layer and parameter designation. (b) Ablation study to investigate the accuracy change as each layer is removed from the intact network architecture. (c) Parameter analysis. Validation accuracy according to filter size, dropout rate, and fully-connected layer (FCL) size in the network learning process. In each analysis, parameters except a variable were fixed at 7 (filter size 1), 5 (filter size 2), 0.4 (dropout rate), 512 (FCL size 1), and 1 (FCL size 2).” - SI p. 11

“For a better understanding of the behavior of the implemented model, we performed additional examinations of the implemented model. Firstly, to compare our model with other machine learning approaches, a dummy classifier and support vector machine (SVM) classifier were brought. In the prediction of the test dataset, our CNN-based model offers superior prediction compared to these two baseline classifiers (Supplementary Fig. 7). In addition, an ablation study was introduced to investigate the contribution of the layer to prediction; thus, we sequentially removed the layer parts of the model and monitored the performance (Supplementary Fig. 8a and 8b). Although overall convolution networks contributed to the performance, the first convolution layer was particularly influential (Supplementary Fig. 8b). Because the first layer is generally associated with basic feature detection of the input data and lessens unwanted features, the result indicates that this process is consistently important in Raman spectrum analysis as well^{42, 43}. In addition, we investigated the influence of structural parameters (Supplementary Fig. 8c). The filter size of the first layer is related to the window size skimming an input spectrum, and performance degrades above about filter size of 10. The filter size of the other layer was saturated above 5. The dropout

rate was consistent below 0.5 and the size of the FCL layer showed no significant effect on performance.” – p. 10-11 Lines 185-200

“Supplementary Fig. 10: Comparison with other models in TOO detection. As a baseline model, dummy classifier and support vector machine (SVM) were established. The accuracy represents data-wise prediction accuracy for the test dataset in each model. In cases where data were overly biased, percentages were displayed.” - SI p. 13

“As with cancer presence detection, we compared the TOO discrimination performance against baseline classifiers (Supplementary fig. 10). The dummy classifier poorly predicted the target class in all TOO discriminant cases, and the SVM classifier showed completely biased results in certain cancer types (breast and pancreatic cancer) as well. These results support that CNN-based classifiers have better performance not only in cancer presence but also in TOO discrimination.” – p. 13 Lines 226-230

To be compatible with Nature Communications, a high degree of scientific rigors and reproducibility is expected. Therefore, source code and data should be opened on GitHub / GitLab / CodeOcean.

- ***Response:*** We agree with the reviewer's opinion. Therefore, we have added the data and code availability section for the guideline and rigors of Nature journals, and disclose source data and codes for figure and result reproduction. All source data constituting the graph in this study are attached to this paper as an Excel file. In addition, the custom codes to draw the figure and sample data for the demo are now provided through the GitHub repository (https://github.com/Hyunkushin/OTMC_NatComm). Through these codes, score prediction using the implemented models will be possible, as well as the reproduction of the main result.

Added manuscript:

“Data availability

Source data constituting the graph in this study are provided as an Excel file with this paper.”

– p. 22 Lines 373-734

“Code availability

All the code used for prediction using implemented algorithms and generation of figure data is available from the GitHub repository (https://github.com/Hyunkushin/OTMC_NatComm) with sample data for the demo.”

– p. 22 Lines 375-377

Right now, the proposed approach looks like a black-box. It is also important to uncover how the proposed approach made decisions from the pathology perspective. Feature importance analysis with WikiPathways may help on deciphering what the proposed approach has learned from SERS, in relation to the AUROCs. On the other hand, the AUPRCs of PRC curves should also be provided.

- ***Response:*** We thank to the reviewer’s point. As suggested by the reviewer, our method relies on algorithm-based classification such as black-box. Exploration of how these classifications are made (i.e., which Raman features affect classification), and correlation with biological databases such as WikiPathways, would provide important implications. However, it is still difficult to explain the direct connection between the Raman signals of human exosomes and biological pathways. More detailed investigations are required now. For example, it may require explainable AI or challenging biological comparative experiments. We are working on these attempts, but unfortunately, this is quite a difficult topic to cover in this paper. Therefore, we thought it worth mentioning these limitations in the discussion section.

Additionally, as pointed out by the reviewer, additional analyzes were performed on the PRC curves and reflected in the main figure and text. As a result, we were able to show through the PRC that our approach exhibits excellent performance even in the detection of positive samples.

Original manuscript ⇒ Revised manuscript

“Nevertheless, several limitations remain to be addressed: first, diagnostic performance must be verified using external tests and prospective clinical trials; second, potential confounding factors, including those generated by benign tumors, should be considered; and third, the biological factors contributing to classification require identification. Ultimately, it will be essential to establish a well-controlled diagnostic process using the actual clinical workflow from blood draw to diagnostic report.” – p. 15 Line 235-240

⇒ “*Nevertheless, several limitations remain to be addressed: first, there is still a need to augment a larger number of training samples, and diagnostic performance must be verified using external tests and prospective clinical trials; second, potential confounding factors, including those generated by benign tumors, should be considered; third, major spectral features should be examined, and its association with reported biological pathways or factors from the pathology perspective; fourth, the introduction of uniform and mass-manufacturable SERS detection chips may be required for precision and reproducibility in clinical practice. Ultimately, it will be essential to establish a well-controlled diagnostic process using the actual clinical workflow from blood draw to diagnostic report.*” – p. 18 Lines 300-308

“Receiver operating characteristic (ROC) curves were calculated to evaluate diagnostic performance. For all HCs and cancer patients, the area under the curve (AUC) value was 0.983 [95% confidence interval (CI) 0.973-0.993] (Fig. 3e). Sensitivity and specificity at the optimal cutoff were 95.5%. AUC values by cancer type were 0.968, 0.999, 0.989, 0.997, 0.988, and 1.00 for lung, breast, colorectal, liver, pancreatic, and stomach cancer, respectively (Fig. 3f). In summary, our system gave accurate predictions for 95.5% (276/289) of all HC and cancer patients. Notably, our system minimized false-positives and false-negatives (Table 1). In 99% specificity condition minimizing false-positive diagnoses, the system had a sensitivity of 77.7%. In 99% sensitivity condition minimizing false-negative patients, the system had a specificity of 71.0%, suggesting that our approach enables the precise identification of multiple types of cancer.” – p. 9-10 Line 151-160

⇒ “Receiver operating characteristic (ROC) curves were calculated to evaluate diagnostic performance. For all HCs and cancer patients, the area under the curve (AUC) value was 0.970 [95% confidence interval (CI) 0.957-0.982] (Fig. 3e). Sensitivity and specificity at the optimal cutoff were 89.4% and 96.3%, respectively. AUC values by cancer type were 0.936, 0.984, 0.972, 0.978, 0.992, and 0.999 for lung, breast, colorectal, liver, pancreatic, and stomach cancer, respectively (Table 1). The F1 score of the precision-recall curve (PRC) and the area under the PRC (AUPRC) value show that the model well predicts positive samples (Fig. 3f). Notably, the model retained effectiveness even under conditions of extreme specificity (Table 1). In 99% specificity condition minimizing false-positive diagnoses, the system had a sensitivity of 72.5%, suggesting that our approach enables the precise identification of multiple types of cancer.” – p. 10 Lines 175-184

“Fig. 3: Cancer presence detection. ... f, PRC curve for the entire test set.”

II. Reviewer #2's comments

II-0

The work of Shin et al. is very interesting and well-designed. In the last years, extracellular vesicles, with emphasis on exosomes (EXOs), are arousing huge interest in cancer diagnosis and vibrational spectroscopies, such as Raman spectroscopy, may be a promising tool for a label-free characterization of these vesicles.

In this work, the Authors exploit surface-enhanced Raman spectroscopy (SERS) for the spectral profiling of EXOs. EXOs were isolated from the serum of a large cohort of healthy donors and cancer patients affected with different types of cancer.

Efficient isolation, and characterization, of EXOs, is a crucial step in works focused on extracellular vesicles. In this case, the Authors used size exclusion chromatography for the EXO isolation, without chemical treatments and chance to introduce artefacts in the samples, as it may happen by using other isolation techniques, such as precipitation kits. In accordance with MISEV 2018 guidelines (<https://doi.org/10.1080/20013078.2018.1535750>), the Authors performed an appropriate characterization of EXOs by western blotting, TEM and NTA.

SERS signals from EXOs were analysed by a machine learning approach providing high diagnostic accuracy and tissue of origin discrimination performance.

This work may pave the way for the development of novel diagnostic techniques by a comprehensive and label-free EXO characterization. Then, I strongly support the publication of this work.

- **Response:** We appreciate the time and effort for review. Based on the reviewer's advice, we added references and text that strengthen the main logic, and tried to address the reviewer's concerns, including a number of samples. Please check the following point-by-point response for this.

The authors should highlight the improvement of their work with respect to the current literature. Are they the first to use Raman spectroscopy for EXO spectral profiling in a such large cohort of cancer patients? Have been differences between the Raman spectra of cancer and non-cancer EXOs (e.g., EXOs from cancer and normal cells) already demonstrated? Please refer to <https://doi.org/10.3390/nano11061476>.

- **Response:** Thank you for the comment. Following the reviewer's advice, we decided to augment the description of Raman spectroscopy for exosomes with additional references below. Based on insights from the references provided, a brief introduction to vibrational spectroscopy and a brief summary of previous Raman spectroscopic applications are now added in the introduction section.

Original manuscript ⇒ **Revised manuscript**

“In this context, Raman spectroscopy is a powerful tool for detecting and analyzing exosomal patterns because it detects vibrational modes related to chemical structures and in turn, allows the identification of compositional differences in biomaterials²⁰⁻²³. With the development of plasmonic enhancing methods, for example surface-enhanced Raman spectroscopy (SERS), many groups have used Raman spectroscopy to analyze EVs^{21, 24-29}. These studies demonstrate that exosome identification is possible without labeling and without specific antigens. In particular, with advances in artificial intelligence (AI) and machine-learning technologies, identification of Raman signals has become more precise^{30, 31}.” – p. 3-4 Lines 53-60

⇒ *“In this context, vibrational spectroscopy techniques that obtain information on vibrational and rotational modes of the chemical structure are emerging as major tools to identify different types of bio-samples²⁰. In particular, Raman spectroscopy is a powerful method for detecting compositional differences in biomaterials such as exosomes based on the advantages of being simple, non-destructive, and requiring less amount of analytes²¹⁻²⁴. With the development of plasmonic enhancing methods, for example surface-enhanced Raman spectroscopy (SERS), many groups have used Raman spectroscopy to analyze EVs^{20, 22, 25-30}. These studies demonstrate that EVs and exosomes from various biosamples from cell-cultured media^{27, 31} to blood^{32, 33} can be identified without labeling and specific antigens. In particular, current advances in multivariate statistical methods and machine-learning technologies, including artificial intelligence (AI), make identifying these vibrational spectra more precise and easy^{20, 31, 33, 34}.” – p. 3-4 Lines 54-64*

Added references:

“Yan, Z. et al. A label-free platform for identification of exosomes from different sources. *ACS sensors* **4**, 488-497 (2019).”

“Romanō, S. et al. Label-free spectroscopic characterization of exosomes reveals cancer cell differentiation. *Analytica Chimica Acta* **1192**, 339359 (2022).”

“Di Santo, R. et al. Recent advances in the label-free characterization of exosomes for cancer liquid biopsy: from scattering and spectroscopy to nanoindentation and nanodevices. *Nanomaterials* **11**, 1476 (2021).”

II-2

The use of artificial intelligence for the development of EXO-based diagnostic techniques is extremely interesting and promising. The Authors should check if some other works combined vibrational spectroscopies, Raman or FTIR, and machine learning-assisted analysis of circulating extracellular vesicles for cancer diagnosis.

- **Response:** Thank you for the comment. As described in response to comment (II-1), we have added a description and references to other works about cancerous exosome and vibrational spectroscopies. Also, we have added several references that describe vibrational spectroscopy-exosome-multivariate statistics. Besides, as described in response to comment I-1, we have added some description about other SERS applications for cancer diagnosis and our novelty, in the discussion section. Please see the previous response to comment (I-1) to find the major changes.

Original manuscript ⇒ **Revised manuscript**

“In this context, Raman spectroscopy is a powerful tool for detecting and analyzing exosomal patterns because it detects vibrational modes related to chemical structures and in turn, allows the identification of compositional differences in biomaterials²⁰⁻²³. With the development of plasmonic enhancing methods, for example surface-enhanced Raman spectroscopy (SERS), many groups have used Raman spectroscopy to analyze EVs^{21, 24-29}. These studies demonstrate that exosome identification is possible without labeling and without specific antigens. In particular, with advances in artificial intelligence (AI) and machine-learning technologies, identification of Raman signals has become more precise^{30, 31}.” – p. 3-4 Lines 53-60

⇒ *“... These studies demonstrate that EVs and exosomes from various biosamples from cell-cultured media^{27, 31} to blood^{32, 33} can be identified without labeling and specific antigens. In particular, current advances in multivariate statistical methods and machine-learning technologies, including artificial intelligence (AI), make identifying these vibrational spectra more precise and easy^{20, 31, 33, 34}.” – p. 4 Lines 60-64*

II-3

Check the numbering of supplementary figures. Supplementary figure 8 should be Supplementary figure 1, Supplementary figure 9 should be Supplementary figure 2, and so on.

- **Response:** We apologized the error of figure numbering. We have now fixed this issue. Thanks for pointing out this mistake.

Samples of EXOs isolated from serum include not only cancer-derived EXOs but vesicles released by blood cells. Have the authors considered the presence of non-cancer EXOs and how they possibly affect the analysis?

- **Response:** We agree with the reviewer’s opinion. As our approach is based on analyzing signals from random exosomes in plasma without selection, some signals may not reflect sample characteristics. In other words, some signals may be indistinguishable because of common exosomes derived from normal cells, even in cancer patient exosomes. Thus, we scanned 100 signals from one sample to capture the characteristic signal of cancerous exosomes as much as possible. Then, in order to draw a single numerical value for the diagnostic criteria, the means of the output values were set as the diagnostic scores for individual samples. The original manuscript lacked explanation for this. Based on the reviewer's comments, these explanations have been added to the manuscript.

Original manuscript ⇒ Revised manuscript

“The first step was to assess whether these data could be used to detect the presence or absence of cancer. To train our model for this purpose, 489 samples were split into training (n = 200) and test (n = 289) samples (Fig. 3a). A total of 19,713 signals (3,945 HC and 15,767 cancer samples) passed anomaly data filtering and were used for training. As our approach is based on analyzing signals from all exosomes in plasma without selection, some signals may not reflect sample characteristics. Therefore, we utilized weakly-supervised learning using inexact labeling of data, such as MIL. That is, by applying the MIL concept, data were collectively labeled with 0 for the control group and 1 for the patient group. The model architecture was composed of serial convolutional layers to conduct binary classification through a sigmoid activation function (Supplementary Fig. 4). Twenty percent of the training dataset was used as a validation set; loss and accuracy in the iteration step are shown in Supplementary Fig. 5.” – p. 8 Lines 124-134

⇒ *“Our approach is to recognize cancer patient samples by machine learning without specifying the characteristic band of the cancerous exosomes. The first step was to assess whether these data could be used to detect the presence or absence of cancer. As our approach is based on analyzing signals from random exosomes in plasma without selection, some signals may not reflect sample characteristics. In other words, some signals may be indistinguishable because of common exosomes derived from normal cells, even in cancer patient exosomes. Thus, we scanned 100 signals from one sample to capture the characteristic signal of cancerous exosomes as much as possible (see Fig. 1a). Then, by applying the multiple instance learning (MIL) concept, an individual spectrum was collectively labeled with 0 for the control group and 1 for the patient group (Fig. 3a), then the average of predicted output derived from an individual sample was used as a single numerical value for diagnostic criteria. The neural network to implement the MIL was composed of a serial convolutional neural network (Supplementary Fig. 5a).” – p. 9 Lines 147-158*

II-5

In line 182 the Authors state that “the system had robust performance in identifying early-stage cancers (Fig. 4c)”. Despite the sample size of the cancer patients being very high, in this case, the number of liver and stomach cancer patients at early-stage is very low (3 and 7 respectively). Please take this into account.

- ***Response:*** We agree with the reviewer's opinion. As mentioned in the previous response to comment (I-2), we augmented the total number of subjects, including early-stage cancer patients, from 289 to 520. Thus, the sample sizes of cancer patients that the reviewer mentioned are now increased (N: Liver 3 → 16, Stomach 7 → 24) The details about this augmentation of samples and changes, please see the previous comment (I-2)

III. Reviewer #3's comments

III-0

The presented manuscript deals with a very promising and interesting field, that is liquid biopsy. The results are well presented and in general the proposed approach sounds technically interesting and biologically challenging. Indeed, exosomes are very interesting cellular vesicles supposed to be potential important actors in several biological pathways. Above all their capability to travel in the plasma give them a very important role in cellular signalling at large distances.

The application of AI algorithms is quite convincing and looks a robust method for analysing spectral data, in order to generate/identify specific disease (cancer) patterns.

The use of SERS technique could seem of crucial importance for achieving a reasonable signal level from exosomes samples, but this reviewer has several concerns about the accomplishment of SERS detection.

Hence, there is enough significance in the achieved results, but some major revisions are required.

- **Response:** We appreciate the time and effort for review. In particular, we are grateful for the detailed review related to SERS, and supplemented the data to address concerns related to SERS detection considering the reviewer's comments. Please check the following point-by-point response for this.

III-1

Lines 110-112 and Figure2f:

The representative spectra look much more like standard Raman spectra than enhanced spectra. Enhanced spectra by SERS are usually characterized by sharper peaks, or at least by some sharper peaks. Despite of the average process, 10 items for the average should leave some sharper features observable in the spectra (due to higher SERS intensities). Usually, SERS spectra require quite low laser powers, in the 0.1-0.5mW range. The laser power of 2mW (reported in materials and methods) is not very high, but also not so low for exciting truly SERS signal. How the author could be sure that the SERS substrate is properly working and providing enhanced spectra, instead of conventional, non-enhanced spectra? and what about the enhancing factors? are they evenly distributed on the substrates or are they rather non-uniform?

- ***Response:*** Thanks for the comment. As the reviewer pointed out, we added the standard deviation range of the Raman spectra to show the sharp peaks of the individual SERS signals. In addition, I agree with the reviewer's opinion about laser power. SERS effect can be induced at lower power and excessive lasers may be unnecessary. Generally, this relies on the properties of the analytes and the SERS substrate. Since our analyte is a biomolecule, it is difficult to detect signals unlike Raman dyes that have chromophores in their chemical structure. (<https://doi-org-ssl.oca.korea.ac.kr/10.1021/ac403974n>) In fact, many other papers that applied label-free SERS to biological materials have also used laser powers of about 2 mW or more. (<https://doi.org/10.1021/acsanm.2c02392>) Thus, we set the conditions to obtain the strongest SERS signal as much as possible while signal fluctuations do not occur. Nevertheless, we agree with the reviewer's concerns about the SERS substrate and have added experimental results for signal differences in SERS and non-SERS conditions to demonstrate the SERS effect and signal enhancement of our substrate. Also, since our substrate is fabricated from colloidal nanoparticles, it is difficult to say that it is uniform from a morphology perspective. For practical application and mass production for clinical use, we agreed on the need for improvement of these SERS substrates and added them as limitations.

“Fig. 2: Exosome isolation and detection. a-d, Evaluation of isolated exosomes. ... f, Representative SERS spectra. Each signal represents a mean of 10 spectra from one sample to show general signal patterns.”

“Fig. 2: Exosome isolation and detection. a-d, Evaluation of isolated exosomes. ... f, Representative SERS spectra. Each graph represents a mean of 10 spectra from one sample to show general signal patterns. The light-colored area represents the standard deviation range.”

“An Au nanoparticle (AuNP)-aggregated array chip for SERS was prepared using centrifugation-based sedimentation methods that we have previously reported³⁴ (Supplementary Fig. 2). After colloidal AuNPs were precipitated, NPs were applied to the APTES-functionalized glass surface as 2.5-mm diameter dots (Fig. 2e). ...” - p. 6 Lines 99-102

⇒ “An Au nanoparticle (AuNP)-aggregated array chip for SERS was prepared using centrifugation-based sedimentation methods that we have previously reported³⁷ (Supplementary Fig. 2). After colloidal AuNPs were precipitated, NPs were applied to the APTES-functionalized glass surface as 2.5-mm diameter dots (Fig. 2e). One chip is designed to have 10 detection spots to increase detection throughput. To evaluate the SERS effect, signals of R6G solution were detected at conditions of SERS and spontaneous Raman (Supplementary Fig. 3a). The enhancement factor was calculated as 4.28×10^5” – p. 6-7 Lines 108-114

⇒ (Added)

“Supplementary Fig. 3: SERS signal enhancement and uniformity. (a) Signal enhancement of the SERS substrate. Each graph represents an averaged spectra of rhodamine 6G (R6G) solution obtained at three different spots. (b) SERS signal uniformity. The SERS signals were scanned after 1 μM of R6G solution was dropped on the array. (c) Intensity maps at 1364 cm⁻¹ of the characteristic bands of R6G from 4 repetitive tests. (d) SEM images of the SERS substrate.”

“Nevertheless, several limitations remain to be addressed: first, diagnostic performance must be verified using external tests and prospective clinical trials; second, potential confounding factors, including those generated by benign tumors, should be considered; and third, the biological factors contributing to classification require identification. Ultimately, it will be essential to establish a well-controlled diagnostic process using the actual clinical workflow from blood draw to diagnostic report.” – p. 15 Lines 235-240

⇒ “Nevertheless, several limitations remain to be addressed: first, there is still a need to augment a larger number of training samples, and diagnostic performance must be verified using external tests and prospective clinical trials; second, potential confounding factors, including those generated by benign tumors, should be considered; third, major spectral features should be examined, and its association with reported biological pathways or factors from the pathology perspective; fourth, the introduction of uniform and mass-manufacturable SERS detection chips may be required for precision and reproducibility in clinical practice. Ultimately, it will be essential to establish a well-controlled diagnostic process using the actual clinical workflow from blood draw to diagnostic report.” – p. 18 Lines 300-308

Lines 164-166 and Supplementary Figure6:

The authors claim that: “We found that true-positive signals tended to deviate from the normal cluster of true-negative signals and were located on the positive side of the principal component 2 (PC2) axis.” This is only partly true. From the figure one can say rather the opposite: that true positive signals are covering the whole range (positive and negative PC2 scores as well as PC1 positive and negative scores), while the true negative signals (HC samples) are only in the negative PC2 range. But there is a large area where HC sample and Cancer samples are overlapping. The authors should re-arrange the statements/results of PCA in less resolute sentences since these results are quite weak from a spectral point of view.

- **Response:** We agree with the author's opinion. PCA analysis was introduced to show the main signal differences between HC and patients exosomes, complementing black box-like algorithms. However, based on the reviewers' comments (including IV-5), this type of analysis was not considered appropriate. Thus, we decided to remove this description and supplementary figure. However, exploring the features that influence classification is still an important factor. Therefore, we have added this as a limitation in the discussion section.

Original manuscript ⇒ **Revised manuscript**

“To identify which signal bands affected this classification, we performed principal component analysis (PCA). First, we collected true-positive and true-negative signals from the training data to identify primary spectral differences (Supplementary Fig. 6). We found that true-positive signals tended to deviate from the normal cluster of true-negative signals and were located on the positive side of the principal component 2 (PC2) axis. In the PC2 loadings, we found several spectral regions that correlated with cancer sample data. Based on the band assignment, several regions were considered to be derived from biological components of EVs, such as proteins and membrane lipids^{30, 37}. These results suggest that the cancer signals identified by our algorithm are derived from differences in exosome composition that are distinct from normal exosomes.”

– p. 10 Lines 161-169

⇒ (Removed)

“Nevertheless, several limitations remain to be addressed: first, diagnostic performance must be verified using external tests and prospective clinical trials; second, potential confounding factors, including those generated by benign tumors, should be considered; and third, the biological factors contributing to classification require identification. Ultimately, it will be essential to establish a well-controlled diagnostic process using the actual clinical workflow from blood draw to diagnostic report.” – p. 15 Lines 235-240

⇒ *“Nevertheless, several limitations remain to be addressed: first, there is still a need to augment a larger number of training samples, and diagnostic performance must be verified using external tests and prospective clinical trials; second, potential confounding factors, including those generated by benign tumors, should be considered; third, major spectral features should be examined, and its association with reported biological pathways or factors from the pathology perspective; fourth, the introduction of uniform and mass-manufacturable SERS detection chips may be required for precision and reproducibility in clinical practice. Ultimately, it will be*

essential to establish a well-controlled diagnostic process using the actual clinical workflow from blood draw to diagnostic report.” – p. 18 Lines 300-308

III-3

This reviewer did not find any info online about the reported exosome isolation kit “Exo-I S5 kit (Exopert)”: should be provided a direct link to this product? how is working this kit? are precipitation, isolation, buffer solutions involved? have been they tested for Raman spectroscopy? It is worthy to point out that every possible source of interference with the spectroscopic results should be addressed and mentioned in the manuscript.

- **Response:** We apologize for missing information related to details about the isolation method. We isolated exosomes by size-exclusion chromatography (SEC). (Product link: http://exopert.co.kr/service/exo_i; http://exopert.co.kr/upload/exo-i%20S5%20manual_221214.pdf) For deeper understanding about this kit, references and brief protocols of papers explaining the principles of this column have been added to the method section. In addition, the SEC method separates substances based on the hydrodynamic size of the particles without chemical conjugation or treatment. Therefore, there will be no additives capable of interfering the Raman signal by the separation process. Nevertheless, we have added a description of possible sources of disturbing signals in the isolation process, following the reviewer's advice.

Original manuscript ⇒ Revised manuscript

“Exosome isolation Frozen suspensions were thawed at 4 °C and 0.5 mL of the resulting plasma was loaded onto a size-exclusion chromatography column using the Exo-I S5 kit (Exopert). PBS was used as the elution buffer; 500-μL of the eluted fractions were collected and evaluated. Isolated exosomes were stored at -80 °C until analysis.” – p. 16 Lines 258-262

⇒ *“Exosome isolation Frozen suspensions were thawed at 4 °C. The exosome isolation was performed using a size-exclusion chromatography column (Exo-I S5, Exopert, KR)³⁶. After substitution of inner liquid, 500-μL of plasma was loaded onto the prepared column. When the plasma is permeated into the column thoroughly, PBS was added as a mobile phase. Then, 500-μL of the eluted fractions were collected serially. Fractions as the exosome suspension was selected after evaluation of collected particles. The resulting suspensions were stored at -80 °C until subsequent analysis.”*
– p. 20 Lines 327-333

“Exosome purity and surface chemical status are significant factors in our label-free detection because other biomolecules and chemicals on the exosome surface affect the resulting Raman signals. Accordingly, we employed size exclusion chromatography (SEC) that isolates exosomes without chemical treatment of the samples^{32, 33}. Firstly, plasma samples were prepared and stored by routine protocols in each medical center. Then, we isolated exosomes from approximately 500-μL plasma samples from 150 healthy controls (HC) and 339 cancer patients whose diagnoses had been pathologically confirmed.” – p. 6 Lines 84-90

⇒ *“Exosome purity and surface chemical status are significant factors in our label-free detection because other biomolecules and chemicals on the exosome surface affect the resulting Raman signals. Accordingly, we employed size exclusion chromatography (SEC) that isolates exosomes based on the hydrodynamic size of vesicles^{35, 36}. Since SEC does not use additional chemical reagents that produce undesired Raman signals in the isolation process, the disturbance of signals can be minimized in label-free SERS detection. Plasma samples from 210 healthy*

controls (HC) and 543 cancer patients had pathologically confirmed their diagnoses by each medical center and stored by routine protocols in each medical center.” – p. 6 Lines 89-96

Added reference:

*“Guo, J. et al. Establishment of a simplified dichotomic size-exclusion chromatography for isolating extracellular vesicles toward clinical applications. Journal of extracellular vesicles **10**, e12145 (2021)”*

*“Jung, J.-H. et al. Dual size-exclusion chromatography for efficient isolation of extracellular vesicles from bone marrow derived human plasma. Scientific reports **11**, 1-9 (2021).”*

III-4

In the Materials and Methods section, the authors mention the Au NP size: “100 nm AuNP colloidal solution (NanoComposix) was concentrated 5-fold through centrifugation...”

100nm is quite a large size to achieve SERS effect directly from the single Au-NPs, so it is reasonable that the gaps between NPs (expected to be in the 10nm range) could be responsible of Raman enhancements. But in this case the NPs layer geometry plays a crucial role. Do the author have an idea of NPs arrangement? in the inset of fig. 2e, the SEM image of NPs shows a random arrangement that could lead to different enhancing factors randomly distributed over the SERS chip. Could the author comment a little bit more on the enhancing properties of the chip? and about their uniformity over the chip area?

The figure “SERS substrate signal uniformity” in the Supplementary shows a quite uniform signal from rhodamine6g, but again the spectra do not own the specific features of SERS signals, but rather they look like standard Raman spectra of rhodamine6g. And again, there is the doubt that the acquired spectra are effectively enhanced Raman signals.

- **Response:** We thank you for the comment. As described in the previous comment (III-1), since our substrate is fabricated from colloidal nanoparticles, it is difficult to say that it is uniform from the morphology perspective. However, the Raman data scanned at different locations showed a relatively uniform signal pattern. It is believed that this is because the signals by the average enhancement of the particles in the laser focused spot and acquisition area ($\sim 1 \mu\text{m}$) were detected. Considering the reviewer’s point, we have added a complementary description of the nanoparticle arrangement, uniformity, and enhancement factor to the manuscript.

Original manuscript \Rightarrow **Revised manuscript**

“An Au nanoparticle (AuNP)-aggregated array chip for SERS was prepared using centrifugation-based sedimentation methods that we have previously reported³⁴ (Supplementary Fig. 2). After colloidal AuNPs were precipitated, NPs were applied to the APTES-functionalized glass surface as 2.5-mm diameter dots (Fig. 2e). To increase detection throughput and acquisition uniformity, a 10-dot array was implemented on the substrate. Signals were automatically measured for each spot using customized software that controls the Raman microscope system. To establish signal acquisition uniformity, we evaluated the trend of 100 signals scanned in the dot using R6G solution (Supplementary Fig. 3). These signals exhibited an identical tendency in intensity at the characteristic band (1364 cm^{-1}) of R6G with an averaged coefficient of variation (CV) of 6.0%. Afterward, the isolated exosomes were dropped onto each dot array and their signals were scanned.”
– p. 6-7 Lines 99-108

\Rightarrow *“An Au nanoparticle (AuNP)-aggregated array chip for SERS was prepared using centrifugation-based sedimentation methods that we have previously reported³⁷ (Supplementary Fig. 2). After colloidal AuNPs were precipitated, NPs were applied to the APTES-functionalized glass surface as 2.5-mm diameter dots (Fig. 2e). One chip is designed to have 10 detection spots to increase detection throughput. To evaluate the SERS effect, signals of R6G solution were detected at conditions of SERS and spontaneous Raman (Supplementary Fig. 3a). The enhancement factor was calculated as 4.28×10^5 . The uniformity in the signal acquisition was*

evaluated through the trend of 100 R6G signals scanned in the dot (Supplementary Fig. 3b). Since our substrate was fabricated based on colloidal nanoparticles, it has a non-uniform hotspot arrangement. Even in this non-uniform condition, relatively uniform signal pattern was observed (Supplementary Fig. 3c). The scanned signals exhibited an identical tendency in intensity at the characteristic band (1364 cm^{-1}) of R6G, showing an averaged coefficient of variation (CV) of 6.0%. This uniformity may be due to the average signal enhancement of the nanoparticles in the laser-focused spot. In this circumstance, the number of the nanoparticles is related to signal enhancement and their average number of particles was 51 ± 6 particles/ μm^2 in SEM characterization (Supplementary Fig. 3d).” – p. 6-7 Lines 108-123

III-5

As already mentioned above, the reported laser power is slightly too high for achieving genuine SERS signals rather than simply conventional Raman spectra. Did the author consider the case that they are recording conventional Raman signals? do they have recorded some comparison spectra (without SERS) to show the effect of the enhancement?

- ***Response:*** Thank you for the comment. As mentioned in the previous answer (III-1), we detected Raman signals of biomaterials, which generally have weak Raman activity. In this setup, many research groups have investigated laser powers similar to ours. However, it is worthwhile to clarify the SERS effect of our setup. Therefore, as mentioned in the previous comment (III-1), the enhancement factor was quantified and presented through a comparison between non-SERE and SERS conditions. Please see the previous response to comment (III-1) to check revised and added data.

III-6

In the Conclusion section: Could the authors comment on the reusability of the device? Is it reusable after proper washing, or is it intended for single use (disposable device)?

- ***Response:*** We thanks for the comment. In our setup, it is very challenging to completely eliminate binding between analytes and nanoparticles; thus, the chips for signal detection are generally not reusable. Considering the reviewer’s comment, we mentioned this point at the end of the discussion.

Original manuscript ⇒ Revised manuscript

“Our approach provides multiple advantages. First, it is rapid. Because we utilized an automated system, exosome isolation (20 min) and detection (30 min for specimen preparation; 10 min for detection) can be completed in an hour. The final decision can be completed in several seconds using pre-trained AI models.... – p. 14 Lines 224-227

⇒ *“As a diagnostic method, our approach provides additional advantages. First, it is rapid. Because we utilized an automated system, exosome isolation (20 min) and detection (30 min for specimen preparation; 10 min for detection) can be completed in an hour. The final decision can be completed in several seconds using pre-trained AI models. The SERS chip with 10 measurement spots is not reusable, but it would be possible for automated detection and diagnosis through programmed stage control. ... – p. 17-18 Lines 287-292*

- **Response:** We apologize for the missing definition for acronym. Currently, if we have modified this part, it has been reflected in the text as follows.

Original manuscript ⇒ **Revised manuscript**

“Fig. 1: One test-multi cancer using exosome-SERS-AI. a, Overview. Exosome suspension is dropped onto an Au nanoparticle-aggregated array chip and thoroughly dried. Signals were observed at 100 spots (10 x 10) per sample and analyzed by AI algorithms. The system outputs predictions about cancer presence and tissue of origin. A heat map shows actual examples of the predicted results for representative clinical subjects. b, AI framework. In the first step, diagnostic scores are assigned as the mean values of the cancer classifier results. In the second step, signals predicted by the previous cancer classifier are analyzed, then an average score is calculated using six type prediction models.” – p. 5 Fig. 1 caption

- ⇒ *“Fig. 1: One test-multi cancer using exosome-SERS-AI. a, Overview. Exosome suspension is dropped onto an Au nanoparticle-aggregated array chip and thoroughly dried. Signals were observed at 100 spots (10 x 10) per sample and analyzed by AI algorithms. The system outputs predictions about cancer presence and tissue of origin. A heat map shows actual examples of the predicted results for representative clinical subjects. b, AI framework. In the first step, diagnostic scores are assigned as the mean values of the multiple instance learning (MIL)-based cancer classifier results. In the second step, signals predicted by the previous cancer classifier are analyzed, then an average score is calculated using six type prediction models.”*
– p. 5 Fig. 1 caption

“Therefore, we utilized weakly-supervised learning using inexact labeling of data, such as MIL. That is, by applying the MIL concept, data were collectively labeled with 0 for the control group and 1 for the patient group. The model architecture was composed of serial convolutional layers to conduct binary classification through a sigmoid activation function (Supplementary Fig. 4).”
– p. 8 Lines 129-134

- ⇒ *“Thus, we scanned 100 signals from one sample to capture the characteristic signal of cancerous exosomes as much as possible (see Fig. 1a). Then, by applying the multiple instance learning (MIL) concept, an individual spectrum was collectively labeled with 0 for the control group and 1 for the patient group (Fig. 3a), then the average of predicted output derived from an individual sample was used as a single numerical value for diagnostic criteria.”*
– p. 9 Lines 152-158

IV. Reviewer #4's comments

IV-0

The article reports on the use of SERS and AI to classify cancer tissue. The workflow is good and the results are convincing. I believe this makes a strong contribution to the field. I only have minor concerns to be addressed.

- **Response**: We appreciate the time and effort for review. Based on the reviewer's advice, we have modified the text to address the reviewer's concerns, including the number of samples, and to make the main logic clearer. Please check the following point-by-point response for this.

IV-1

I think one of the limitation is the number of samples used for training and for the validation. While I understand that the number of spectra is relatively standard, the number of clinical samples in the case of some cancer is low. How the model holds on with larger number of samples remains to be shown and this need to be highlighted in the limitation section.

- ***Response:*** We agree with the reviewer's opinion. As mentioned in the previous response to comment (I-2), we augmented the total number of subjects, including early-stage cancer patients, from 289 to 520. Thus, the sample sizes of cancer patients that the reviewer mentioned are now increased (N: Liver 3 → 16, Stomach 7 → 24) The details about this augmentation of samples and changes, please see the previous comment (I-2)

IV-2

Related to this, it is not clear how the repeat spectra were collected from samples to reach nearly 20,000 signals. Is it from multiple spectra on a single spot or from multiple samples?

- **Response:** Thank you for pointing out this. We used about 100 signals acquired from different spots on the SERS substrate on which one sample was applied for algorithm training and analysis. As described in previous response to comment (II-4), this approach was intended to capture the characteristic signal of cancerous exosomes as much as possible in a complex and heterogeneous exosome population. The original manuscript lacked explanation for this. For clear understanding, we further commented on these details about our approach in detection in the text:

Original manuscript ⇒ **Revised manuscript**

“The first step was to assess whether these data could be used to detect the presence or absence of cancer. To train our model for this purpose, 489 samples were split into training ($n = 200$) and test ($n = 289$) samples (Fig. 3a). A total of 19,713 signals (3,945 HC and 15,767 cancer samples) passed anomaly data filtering and were used for training. As our approach is based on analyzing signals from all exosomes in plasma without selection, some signals may not reflect sample characteristics. Therefore, we utilized weakly-supervised learning using inexact labeling of data, such as MIL. That is, by applying the MIL concept, data were collectively labeled with 0 for the control group and 1 for the patient group. The model architecture was composed of serial convolutional layers to conduct binary classification through a sigmoid activation function (Supplementary Fig. 4). Twenty percent of the training dataset was used as a validation set; loss and accuracy in the iteration step are shown in Supplementary Fig. 5.” – p. 8 Lines 124-134

⇒ *“Our approach is to recognize cancer patient samples by machine learning without specifying the characteristic band of the cancerous exosomes. The first step was to assess whether these data could be used to detect the presence or absence of cancer. As our approach is based on analyzing signals from random exosomes in plasma without selection, some signals may not reflect sample characteristics. In other words, some signals may be indistinguishable because of common exosomes derived from normal cells, even in cancer patient exosomes. Thus, we scanned 100 signals from one sample to capture the characteristic signal of cancerous exosomes as much as possible (see Fig. 1a). Then, by applying the multiple instance learning (MIL) concept, an individual spectrum was collectively labeled with 0 for the control group and 1 for the patient group (Fig. 3a), then the average of predicted output derived from an individual sample was used as a single numerical value for diagnostic criteria. The neural network to implement the MIL was composed of a serial convolutional neural network (Supplementary Fig. 5a).” – p. 9 Lines 147-158*

The ratio of training to test is unusual, where fewer training samples were used. Could the author explain why?

- Response:** We thank the reviewer for giving us useful comments. We thought it was important to evaluate whether the model worked correctly on a large number of test samples for the final diagnosis. Thus, we investigated how many training samples are needed to predict independent test samples. We evaluated the accuracy on test samples while increasing the number of training samples. As a result, we confirmed that the accuracy tendency saturates stably over 30 to 40 samples per class (cancer and non-cancer). Based on these results, we set 50 for HC and 183 for cancer patients as the number of training samples to implement models. We have added these additional experimental results and information to the text.

Revised manuscript:

⇒ (Added) “First, we investigated how many training samples are needed to predict unknown samples. For this purpose, the accuracy of independent samples was examined while increasing the number of training samples (Supplementary Fig. 5b). As a result, the accuracy tendency was saturated at over 30 ~ 40 samples per class. Based on these results, we set 50 for HC and 183 for cancer patients as the number of training samples to implement models. Accordingly, the entire sample was split into training ($n = 233$) and test ($n = 520$) samples (Fig. 3a). A total of 23,051 signals (4,943 HC and 18,108 cancer samples) passed anomaly data filtering and were used for training.”

– p. 9 Lines159-165

“Supplementary Fig. 5: Deep-learning model. (a) Architecture; f , k , and s indicate the numbers of filters, kernel size, and stride size in each convolution layer (Conv 1D). For binary classification, the final activation function was set to be sigmoid. (b) Gradual change in test accuracy with the number of training samples per class. Three repetitions were conducted through random sampling. (c) Loss and accuracy curves from the implementation of the cancer diagnosis model. The RMSprop optimizer was utilized in the learning step with a learning rate of 0.00035, a decay rate of 0.000181, and a batch size of 32.”

– p. 9 Lines159-165

IV-4

In Fig 2, it would be nice to have the differential spectra from each cancer type in relation to HC. The differences mentioned in lines 111 to 114 would be more evident.

- **Response:** Thanks for providing a useful comment. Following the reviewer's advice, we performed direct Raman signal comparison through the difference spectrum between HC and each cancer type. Accordingly, it was possible to extract peaks information common to cancer exosomes, and a description thereof was added to the text.

Original manuscript ⇒ Revised manuscript

“Fig. 2f shows representative spectra. The common broad and strong signals near 860, 1283, and 1597 cm^{-1} likely indicate citrate molecules on the AuNP surface or protein components such as tyrosine, phenylalanine, tryptophan, and amide III^{34, 35}. We observed subtle peak and intensity variations in several regions near 638, 668, 707, 733, 978, 1001, 1049, 1123, 1162, 1358, 1378, 1394, and 1432 cm^{-1} , which correspond with protein constituents³⁶. To analyze these patterns to extract useful features in cancer patient samples, we applied a deep learning approach.”

– p.7 Lines 109-114

⇒ *“To detect SERS signals of the isolated exosomes, exosomes were dropped onto each dot array and their signals were scanned after thoroughly dried. Fig. 2f shows representative spectra of exosomes isolated from each group. The common broad and strong signals near 860, 1283, and 1597 cm^{-1} likely indicate citrate molecules on the AuNP surface or protein components such as tyrosine, phenylalanine, tryptophan, and amide III^{37, 38}. We observed subtle peak and intensity variations in several regions near 638, 668, 707, 733, 978, 1001, 1049, 1123, 1162, 1358, 1378, 1394, and 1432 cm^{-1} , which can be assigned to protein and lipid constituents^{22, 33, 39, 40}. Difference spectra was investigated to confirm the major difference in Raman signal between HC and cancer patients (Supplementary Fig. 4). As a result, we identified several Raman bands common across all cancer groups near 691, 826, 938, 961, 993, 1136-1152, 1245, 1527, and 1595 cm^{-1} . Most signal bands are assigned to protein constituents⁴¹.”* – p. 7-8 Lines 124-134

“Supplementary Fig. 4: Signal difference. The difference between the average SERS signal of each cancer type and HC is shown. Min-max normalization was performed prior to spectrum subtraction to reduce overall intensity variation. The gray dotted lines indicate common bands detected from all cancer types.” – SI p. 7

IV-5

I don't agree with the statement that PCA identifies the signals used by the ML to classify. Both are different mathematical models and there is a strong chance that different weights are given to different Raman frequencies in the classification. This needs to be better supported or corrected.

- **Response:** We agree with the reviewer's opinion. The PCA can be distinguished from implemented machine learning by other criteria. The PCA analysis was introduced to show the main signal difference between HC and patients exosomes, complementing the black box-like algorithm. However, based on the comments of the reviewers (including III-2), these analysis were no longer considered appropriate. Instead, as mentioned at IV-4, we decided to directly compare and analyze the signal difference between HC and each cancer through difference spectrum. However, exploring the features that influence classification is still an important factor. Therefore, we have added this as a limitation in the discussion section.

Original manuscript ⇒ Revised manuscript

“To identify which signal bands affected this classification, we performed principal component analysis (PCA). First, we collected true-positive and true-negative signals from the training data to identify primary spectral differences (Supplementary Fig. 6). We found that true-positive signals tended to deviate from the normal cluster of true-negative signals and were located on the positive side of the principal component 2 (PC2) axis. In the PC2 loadings, we found several spectral regions that correlated with cancer sample data. Based on the band assignment, several regions were considered to be derived from biological components of EVs, such as proteins and membrane lipids^{30, 37}. These results suggest that the cancer signals identified by our algorithm are derived from differences in exosome composition that are distinct from normal exosomes.”

– p. 10 Lines 161-169

⇒ (Removed)

“Nevertheless, several limitations remain to be addressed: first, diagnostic performance must be verified using external tests and prospective clinical trials; second, potential confounding factors, including those generated by benign tumors, should be considered; and third, the biological factors contributing to classification require identification. Ultimately, it will be essential to establish a well-controlled diagnostic process using the actual clinical workflow from blood draw to diagnostic report.” – p. 15 Lines 235-240

⇒ *“Nevertheless, several limitations remain to be addressed: first, there is still a need to augment a larger number of training samples, and diagnostic performance must be verified using external tests and prospective clinical trials; second, potential confounding factors, including those generated by benign tumors, should be considered; third, major spectral features should be examined, and its association with reported biological pathways or factors from the pathology perspective; fourth, the introduction of uniform and mass-manufacturable SERS detection chips may be required for precision and reproducibility in clinical practice. Ultimately, it will be essential to establish a well-controlled diagnostic process using the actual clinical workflow from blood draw to diagnostic report.” – p. 18 Lines 300-308*

IV-6 Lines 182 to 187, I am worried that the sample number is too low to make such discrimination.

- **Response**: We agree with the reviewer's opinion. As mentioned in the previous response to comment (I-2), we augmented the total number of subjects, including early-stage cancer patients, from 289 to 520. Thus, the sample sizes of cancer patients that the reviewer mentioned are now increased (N: Liver 3 → 16, Stomach 7 → 24) The details about this augmentation of samples and changes, please see the previous comment (I-2)

IV-7 Figure numbering in SI is incorrect. It starts at S8 and there are two Figures S8

- **Response**: We apologized the error of figure numbering. We have now fixed this issue. Thanks for pointing out this mistake.

IV-8 Page numbers are striked in SI.

- **Response**: We apologized the formatting error. We have now fixed this. Thanks for pointing out this mistake.

REVIEWERS' COMMENTS

Reviewer #1 (Remarks to the Author):

The authors have addressed my comments.

Reviewer #2 (Remarks to the Author):

The Authors have addressed all of my concerns. The Authors have increased the sample size of the study and, therefore, also increased its robustness. In my opinion, the manuscript, in this revised version, can be accepted for publication.

Reviewer #3 (Remarks to the Author):

The authors have satisfactorily answered to all questions arisen by this reviewer. The corrections and improvements of both text and figures makes the presented manuscript acceptable for publication.

Reviewer #4 (Remarks to the Author):

The authors addressed all my comments and I recommend publication.

Response to comments

I. Reviewer #1's comments

I-0 The authors have addressed my comments.

- **Response:** We really thank the reviewer for the scientific advice during the whole review process.

II. Reviewer #2's comments

II-0 The Authors have addressed all of my concerns. The Authors have increased the sample size of the study and, therefore, also increased its robustness. In my opinion, the manuscript, in this revised version, can be accepted for publication.

- **Response:** We really thank the reviewer for the scientific advice during the whole review process.

III. Reviewer #3's comments

III-0 The authors have satisfactorily answered to all questions arisen by this reviewer. The corrections and improvements of both text and figures makes the presented manuscript acceptable for publication.

- **Response:** We really thank the reviewer for the scientific advice during the whole review process.

IV. Reviewer #4's comments

IV-0 The authors addressed all my comments and I recommend publication.

- **Response:** We really thank the reviewer for the scientific advice during the whole review process.